# Sweat Proteomics in Cystic Fibrosis: Discovering Companion Biomarkers for Precision Medicine and Therapeutic Development

**DOI:** 10.3390/cells11152358

**Published:** 2022-07-31

**Authors:** Bastien Burat, Audrey Reynaerts, Dominique Baiwir, Maximilien Fléron, Sophie Gohy, Gauthier Eppe, Teresinha Leal, Gabriel Mazzucchelli

**Affiliations:** 1Mass Spectrometry Laboratory, MolSys Research Unit, Liège Université, B-4000 Liège, Belgium; g.eppe@uliege.be; 2Louvain Center for Toxicology and Applied Pharmacology (LTAP), Institut de Recherche Expérimentale et Clinique (IREC), Université Catholique de Louvain, B-1200 Brussels, Belgium; audrey.reynaerts@saintluc.uclouvain.be (A.R.); teresinha.leal@uclouvain.be (T.L.); 3GIGA Proteomics Facility, Liège Université, B-4000 Liège, Belgium; d.baiwir@uliege.be (D.B.); m.fleron@uliege.be (M.F.); 4Cystic Fibrosis Reference Center, Cliniques Universitaires Saint-Luc, Université Catholique de Louvain, B-1200 Brussels, Belgium; sophie.gohy@uclouvain.be; 5Department of Pneumology, Cliniques Universitaires Saint-Luc, Université Catholique de Louvain, B-1200 Brussels, Belgium; 6Pole of Pneumology, ENT and Dermatology, Institut de Recherche Expérimentale et Clinique (IREC), Université Catholique de Louvain, B-1200 Brussels, Belgium

**Keywords:** cystic fibrosis, human eccrine sweat, shotgun proteomics, companion biomarkers, actin cytoskeleton

## Abstract

In clinical routine, the diagnosis of cystic fibrosis (CF) is still challenging regardless of international consensus on diagnosis guidelines and tests. For decades, the classical Gibson and Cooke test measuring sweat chloride concentration has been a keystone, yet, it may provide normal or equivocal results. As of now, despite the combination of sweat testing, *CFTR* genotyping, and CFTR functional testing, a small fraction (1–2%) of inconclusive diagnoses are reported and justifies the search for new CF biomarkers. More importantly, in the context of precision medicine, with a view to early diagnosis, better prognosis, appropriate clinical follow-up, and new therapeutic development, discovering companion biomarkers of CF severity and phenotypic rescue are of utmost interest. To date, previous sweat proteomic studies have already documented disease-specific variations of sweat proteins (e.g., in schizophrenia and tuberculosis). In the current study, sweat samples from 28 healthy control subjects and 14 patients with CF were analyzed by nanoUHPLC-Q-Orbitrap-based shotgun proteomics, to look for CF-associated changes in sweat protein composition and abundance. A total of 1057 proteins were identified and quantified at an individual level, by a shotgun label-free approach. Notwithstanding similar proteome composition, enrichment, and functional annotations, control and CF samples featured distinct quantitative proteome profiles significantly correlated with CF, accounting for the respective inter-individual variabilities of control and CF sweat. All in all: (i) 402 sweat proteins were differentially abundant between controls and patients with CF, (ii) 68 proteins varied in abundance between F508del homozygous patients and patients with another genotype, (iii) 71 proteins were differentially abundant according to the pancreatic function, and iv) 54 proteins changed in abundance depending on the lung function. The functional annotation of pathophysiological biomarkers highlighted eccrine gland cell perturbations in: (i) protein biosynthesis and trafficking, (ii) CFTR proteostasis and membrane stability, and (iii) cell-cell adherence, membrane integrity, and cytoskeleton crosstalk. Cytoskeleton-related biomarkers were of utmost interest because of the consistency between variations observed here in CF sweat and variations previously documented in other CF tissues. From a clinical stance, nine candidate biomarkers of CF diagnosis (CUTA, ARG1, EZR, AGA, FLNA, MAN1A1, MIA3, LFNG, SIAE) and seven candidate biomarkers of CF severity (ARG1, GPT, MDH2, EML4 (F508del homozygous), MGAT1 (pancreatic insufficiency), IGJ, TOLLIP (lung function impairment)) were deemed suitable for further verification.

## 1. Introduction

In clinical routine, the diagnosis of Cystic Fibrosis (CF) is confirmed in the presence of both clinical manifestations of the disease and evidence of CFTR loss-of-function, according to consensus, yet guidelines are still evolving [1,2]. As of today, CF diagnosis remains challenging due to: (i) the high number of mutations and phenotypes [3,4,5], (ii) the struggle with the differential diagnosis in a small fraction (1–2%) of equivocal cases (e.g., between CF and CFTR-Related Metabolic Syndrome/CF Screen Positive, Inconclusive Diagnosis (CRMS/CFSPID) cases), but also in regard to (iii) the worldwide implementation of currently available newborn screening strategies and diagnosis tests [6]. The Gibson and Cooke sweat test [7] (the measurement of chloride concentration in sweat secreted following cholinergic stimulation with pilocarpine) has been the cornerstone of CF diagnosis for decades. Genetic analysis of *CFTR* and functional testing of CFTR (nasal potential difference, intestinal current measurements, or forskolin-induced swelling in intestinal organoids) complete the panel of currently available diagnostic tests. CF likeliness is high for sweat chloride concentrations above 60 mM, in the presence of two CF-causing mutations together with the demonstration of CFTR dysfunction. However, CF is unlikely for sweat chloride concentrations below 30 mM, in the absence of CF-causing mutations together with the demonstration of the functional integrity of CFTR. Cases featuring normal or equivocal sweat chloride concentrations, *CFTR* variants with unknown phenotypic signification and equivocal CFTR function make room for the development of new CF biomarkers. Moreover, in the context of personalized medicine, prognostic biomarkers and companion diagnostic biomarkers are valuable for the earliest and most appropriate therapeutic intervention [8]. In addition, in the context of therapeutic development, biomarkers of clinical response and phenotypic rescue (e.g., CFTR/F508del CFTR physical or functional protein interactors) are critical to the emergence and advance of new therapies [9,10,11].

Sweat is a biological fluid of utmost interest in the search for CF biomarkers since sweat collection and testing have already been integrated into CF diagnosis guidelines. The collection of sweat is less invasive than that of other matrices such as blood and bronchoalveolar fluid or nasal and intestinal biopsies. Moreover, the recent advances in high-resolution, high-sensitivity analytical techniques allow working with sweat volumes collected by means of standardized collection methods [12,13,14,15] on par with the Gibson and Cooke sweat test volume ranges and standards, typically ranging from 20 to 100 µL for a protein concentration ranging from 0.1 to 1 µg·µL^−1^.

Remarkably, despite a low protein concentration, the proteomic analysis of human sweat already proved to be an informative source of pathophysiological biomarkers. For instance, disease-specific profiles of sweat proteins were described in patients with schizophrenia [12] or tuberculosis [13].

Here, with a view to discovering new protein biomarkers of CF for precision medicine and therapeutic development, the proteomic profiles of sweat from patients with CF and healthy control subjects were compared. Downstream sweat collection following the sweat test gold standard, the nanoUHPLC-Q-Orbitrap-based analytical workflow characterized, identified, and quantified 1057 proteins at an individual level, by means of a shotgun label-free strategy. Distinct sweat proteome profiles were observed in CF, due to pathophysiological inter-individual variations of the sweat proteome. Functional annotation of the differentially expressed proteins was performed. Interestingly, cytoskeleton-related biomarkers varied consistently in CF sweat, as reported here, and in other CF tissues, as documented in previous works. With regards to the clinical relevance of sweat proteins, candidate biomarkers of CF diagnosis and CF severity were deemed of interest for further investigation.

## 2. Materials and Methods

The study was conducted according to the guidelines of the Declaration of Helsinki and approved by the Ethics Committee of Cliniques Universitaires Saint-Luc–Université Catholique de Louvain faculty hospital (ClinicalTrials.gov identifier: NCT03993600, date of approval 4 December 2018). A material transfer agreement was signed with the University of Liège for sample analysis. Informed consent was obtained from all subjects involved in the study.

### 2.1. Sweat Collection

Sweat samples were collected from 30 healthy volunteers (15 females, 15 males) and 15 patients with CF (11 males, 4 females) under the most standardized and spectroscopically pure conditions following the current recommendations for the Gibson and Cooke sweat test [16]. In short, the volar region of the forearms was chosen based on its high density in eccrine glands and low density in apocrine/apoeccrine glands together with its easy access. Sweat samples were collected from each forearm successively, from fasting and well-hydrated individuals. Before sampling, the tested region was washed with 70% ethanol, rinsed with ultrapure water, and dried using ashless filter paper. Sweat secretion was stimulated by pilocarpine iontophoresis using pilocarpine gel-padded (Pilogel^®^ discs, ELITechGroup, Brussels, Belgium) electrodes. A 5 mA current (Webster Sweat Inducer, Model 3700, ELITechGroup, Brussels, Belgium) was applied for 5 min. After stimulation, the electrodes were removed and a Macroduct Sweat Collector (ELITechGroup, Brussels, Belgium) (blue dye removed with 70% ethanol and ashless filter paper) was attached in the place of the cathode to collect sweat for 30 min. At the end of the collection, the tubing was uncoiled, cut off, and connected to a needle and syringe to transfer sweat to a 0.6 mL micro-tube.

### 2.2. Sample Preparation for Shotgun Proteomics

Pure, undiluted sweat samples were processed in three series of ten control samples and two series of seven and eight CF samples. Five sample preparation rounds (1 per series) were performed to avoid any technical bias that might come from a single sample preparation experiment.

Sweat protein concentration was estimated using the Pierce Micro BCA™ Protein Assay kit (#23235, ThermoFisher Scientific, Waltham, MA, USA) according to the manufacturer’s instructions. Ten micrograms of proteins were precipitated by incubation in 90% acetonitrile for 30 min at 4 °C followed by centrifugation for 10 min at 4 °C, at 10,000× *g*. The protein pellet was re-suspended in 50 mM ammonium bicarbonate and then incubated in: (i) 10 mM DTT (dithiothreitol) for 40 min at 56 °C, under stirring at 600 rpm (Thermomixer comfort, Eppendorf, Hamburg, Germany), to reduce disulfide bonds, (ii) 20 mM iodoacetamide protected from light for 30 min at room temperature to alkylate/block cysteine residues, (iii) 11 mM DTT protected from light for 10 min at room temperature to quench the residual iodoacetamide, (iv) mass-spectrometry grade trypsin (Pierce™ Trypsin Protease, MS Grade, ThermoFisher Scientific, Waltham, MA, USA) at a 1:50 enzyme:protein ratio (protein concentration = 0.25 µg·µL^−1^) for 18 h at 37 °C, under stirring at 600 rpm (Thermomixer comfort, Eppendorf, Hamburg, Germany), (v) MS-grade trypsin in 80% acetonitrile, at a 1:100 enzyme:protein ratio for 3 h at 37 °C, under stirring at 600 rpm (Thermomixer comfort, Eppendorf, Hamburg, Germany). Digestion was stopped by adding TFA (trifluoroacetic acid) to a final concentration of 0.5% (*v*/*v*). Samples were dried in a vacuum concentrator and re-suspended at 3.75 µg/20 µL in 0.1% TFA. At this step, aliquots from each control sample were collected and mixed in three 10-sample pools, considering three series of 10 individual samples and an average pooled sample per series. CF samples were only processed as individual samples. No CF pooled library was prepared. Individual samples and pooled samples were desalted with C18 Zip Tips according to the manufacturer’s recommendations, dried, and re-suspended at 3 µg/9 µL (injection volume) in 0.1% TFA spiked with an equivalent of MassPREP Digestion Standard Mixture 1 (MPDS Mix 1, #186002865, Waters, Milford, MA, USA) corresponding to 50 fmol of ADH (alcohol dehydrogenase 1 from *Saccharomyces cerevisiae*) content per injection volume.

### 2.3. Liquid Chromatography and Mass Spectrometry Data Acquisition

The control individual samples and pooled samples were randomly sorted into three series of ten individual samples plus one pooled sample while the CF samples were randomly sorted into two series of seven and eight individual samples. All samples were analyzed using an ACQUITY UPLC M-Class liquid chromatography system (Waters, Milford, MA, USA) coupled to a Q-Exactive Plus Hybrid Quadrupole-Orbitrap mass spectrometer (ThermoFisher Scientific, Waltham, MA, USA). Five acquisition rounds (1 per series) were performed to avoid any technical bias that might come from a single LC-MS acquisition series.

The chromatographic separation consisted of a 3 min long trapping step performed on a reversed-phase (RP) ACQUITY UPLC M-Class Trap Column (nanoEase MZ Symmetry C18 Trap Column, 100 Å, 5 μm, 180 μm × 20 mm, Waters, Milford, MA, USA) followed by a 177–minute elution step on an ACQUITY UPLC M-Class Analytical Column (nanoEase MZ HSS T3 C18 Analytical Column, 100 Å, 1.8 μm, 75 μm × 250 mm, Waters, Milford, MA, USA) using a gradient of mixed water and acetonitrile, both supplemented with 0.1% formic acid as eluents.

The mass acquisition was operated in data-dependent positive ion mode. Source parameters were set at: (i) 2.3 kV for spray voltage, (ii) 270 °C for capillary temperature, (iii) S-lens RF level = 50.0.

For individual samples, MS spectra were obtained for scans between *m*/*z* 400 and *m*/*z* 1600 with a mass resolution of 70,000 at *m*/*z* 200, an Automated Gain Control (AGC) of 3 × 10^6^, a maximum Injection Time (IT) of 200 ms, and an internal lock mass calibration at *m*/*z* 445.12003. MS/MS spectra were obtained for the top 10 most intense ions of each MS scan (TopN = 10) with a mass resolution of 17.500 at *m*/*z* 200, an isolation window of 1.6 *m*/*z* with an isolation offset of 0.5 *m*/*z*, an AGC of 1 × 10^5^, a maximum IT of 200 ms, and an (N)CE at 28. The exclusion of single-charged ions and a dynamic exclusion of 10 s were enabled.

For each pooled sample, the MS acquisition consisted of a two-round strategy of three injections each. During both rounds, MS spectra were obtained for scans between *m*/*z* 400 and *m*/*z* 528.3, *m*/*z* 524.3 and *m*/*z* 662.8, or *m*/*z* 658.8 and *m*/*z* 1600, in three independent analyses, respectively, with a mass resolution of 70,000 at *m*/*z* 200, an AGC of 3 × 10^6^, a maximum IT of 200 ms, and internal lock mass calibrations at *m*/*z* 445.12003, *m*/*z* 536.16537, and *m*/*z* 684.20295, respectively. During the first acquisition round, MS/MS spectra were obtained for the top 25 most intense ions of each MS scan (TopN = 25) with a mass resolution of 17.500 at *m*/*z* 200, an isolation window of 1.6 *m*/*z* with an isolation offset of 0.5 *m*/*z*, an AGC of 1 × 10^5^, a maximum IT of 250 ms, and an (N)CE at 28. For the second acquisition round, an exclusion list for all signals related to peptides identified in the first round with more than 4 PSM (peptide–spectrum matches) was uploaded to the methods. During the second acquisition round, MS/MS spectra were obtained for the top 10 most intense ions of each MS scan (TopN = 10) with a mass resolution of 17.500 at *m*/*z* 200, an isolation window of 1.6 *m*/*z* with an isolation offset of 0.5 *m*/*z*, an AGC of 1 × 10^5^, a maximum IT of 600 ms and an (N)CE at 28. The exclusion of single-charged ions and a 15 s dynamic exclusion were enabled for both rounds.

### 2.4. Bioinformatic Analysis

Raw MS data were submitted to protein identification and label-free quantification by the MaxQuant software [17] (version 1.6.6.0) using default settings when not specified otherwise. Identification consisted of a search against a custom-made reviewed Uniprot *Homo sapiens* database (20443 *Homo sapiens* entries + 4 MPDS Mix 1 entries, release date 8 August 2019) with Carbamidomethyl (C) set as a fixed modification Oxidation (M), Deamidation (NQ) set as variable modifications and a minimum of two peptides (including one unique peptide) required. LFQ was enabled and separated between control and CF groups with a minimum LFQ ratio count of 1, no Fast LFQ, and no requirement of MSMS for LFQ comparison. The ‘match between runs’ (MbR) option was enabled and tuned to allow matches from the library (pooled samples considered as parameter group 1, ‘match from’) and between individual samples (parameter groups 0 and 2, ‘match from and to’). A match time window of 2.5 min was used.

MaxQuant output data (proteingroups.txt) were submitted to statistical analysis using the Perseus software [18] (version 1.6.10.43). ‘Only identified by site’, ‘REVERSED’, and Contaminant data were filtered out. LFQ intensities were log2-transformed and proteins with less than 50% of valid values were filtered out. Principal Component Analysis (PCA) was performed on Z-score-normalized LFQ intensities.

Computation of Pearson’s correlation coefficients (PCC) and average Euclidian distance hierarchical clustering were used to classify samples according to quantitative profiles of pairwise correlation with other samples in the cohort, without a priori. Available clinical parameters were tested for the significance of their effects on the proteome profile clustering by PERMANOVA.

PERMANOVA (PERmutational Multivariate ANalysis Of VAriance) tests were performed with the PAST software [19] (version 4.04) using the hierarchical clustering distance matrix with a number of permutations set to 999.

Sparse Partial Least Squares (SPLS) regression was performed as an unsupervised multivariate analysis to test the association between highly correlated covariates from the proteomic data (matrix X of multivariate predictors) and the clinical data (matrix Y of multivariate responses). Before SPLS processing, the predictors and responses were centered. *η*, a sparsity tuning parameter, and *k*, the number of latent components were determined among possible numbers for *η* − *η* ∈ (0.1,0.9) and *k* − 1 ≤ *k* ≤ 15. η=0.69 and *k* = 2 minimized the mean squared prediction error (MSPE, Appendix A).

Control versus CF group comparison was achieved by a two-sample Student’s *t*-test with a *p*-value-based threshold. Proteins with a *p*-value below 0.05 were considered significantly differentially expressed between the control and CF groups. A second comparison was performed by a two-sample Student’s *t*-test with a permutation-based FDR calculation and a q-value-based threshold.

Differentially abundant proteins were characterized based on both the *p*-value (a less stringent cut-off threshold (*p* < 0.05) for biological relevance) and q-value (a more stringent permutation-based FDR threshold (q < 0.05) for clinical biomarker relevance) thresholds. On top of the q-value cut-off, stringent data filtering based on sample occurrence, difference significance, and difference value was applied to a shortlist of candidate clinical biomarkers.

Control versus CF Volcano plot visualization was achieved by a two-sample Student’s *t*-test with a permutation-based FDR calculation (test = *t*-test; side = both; number of randomizations = 250; no grouping in randomizations; FDR = 0.05; s0 = 0.1).

Functional annotations of identified proteins and over-representation/enrichment tests were conducted using the online search engine powered by the PANTHER Classification system. The PANTHER Overrepresentation test (release date: 28 July 2020 and 24 February 2021) parsed the PANTHER database (version 16.0, release date: 1 December 2020) using the *Homo sapiens* reference list, the PANTHER-Gene Ontology-Slim, and the PANTHER Protein Class annotation dataset. Only *p* < 0.05 items were retained and considered significantly over-represented.

Visualization of proteome overlaps was performed by submitting SwissProt accession IDs to the online Venn diagram generator Venny (version 2.1.0, https://bioinfogp.cnb.csic.es/tools/venny, accessed on 8 October 2021).

Visualization of interaction networks was performed by submitting SwissProt accession IDs to STRING (version 11.0, https://string-db.org, accessed on 13 October 2021).

For concision, all protein names were abbreviated with the related gene names. 

### 2.5. Characterization of Sweat Actin

Sweat actin concentration was determined for each individual sample using the “Total Protein Approach” to estimate absolute quantification [20]. A sweat aliquot equivalent to 5 ng of actin was diluted with ultrapure water (q.s. 10 µL, final concentration 0.5 ng·µL^−1^). Phalloidin-TRITC stock solution was prepared by dissolving 0.1 mg of phalloidin-TRITC (P1951, Phalloidin-TRITC peptide from *Amanita phalloides*, Sigma-Aldrich) in 1 mL of 50% methanol (stock concentration 0.1 mg·mL^−1^). Buffer A consisted of a 20 mM potassium acetate/20 mM Tris acetate, pH 7.5 solution. F-actin microfilaments were labeled with phalloidin-TRITC by diluting sweat aliquots with buffer B (1:10 dilution of dye conjugate stock solution in buffer A) to a final concentration of 0.25 ng·µL^−1^ (final volume: 20 µL) and incubating for 45 min, at room temperature, under stirring at 600 rpm, protected from light.

To eliminate residual dye conjugates, labeled F-actin was precipitated with other sweat proteins by incubation in 90% acetonitrile for 30 min at 4 °C followed by centrifugation for 10 min at 4 °C, at 10,000× *g*. The supernatant containing free dye conjugates in the solution was discarded. The protein pellet containing phalloidin-TRITC-F-actin was re-suspended in 20 µL of buffer A. Twenty µL of labeled F-actin solution were put on a microscope slide covered with a cover slip sealed with nail polish.

Ten individual slides were prepared from ten individual sweat samples (5 controls, 5 CFs). F-actin microfilaments were observed with an Olympus IX81 inverted microscope (Olympus Corporation, Tokyo, Japan) equipped with an X-Cite 120PC Q 120 W Hg lamp (Excelitas Technologies, Waltham, MA, USA) for epifluorescence illumination, a red filter (excitation wavelength range 560 +/− 55 nm; emission wavelength range 645 +/− 75 nm) and a 100X oil-immersed objective (UPLSAPO100XO, Olympus Corporation, Tokyo, Japan). The image was detected using an Olympus XC50 CCD color camera (Olympus Corporation, Tokyo, Japan).

For each individual microscope slide, 20 snapshots were taken as followed: 10 snapshots at an operator-chosen position and 10 snapshots at a random position.

Images were automatically analyzed using ImageJ (version 1.53c). Image analysis consisted of an in-house macro implementation: pixel-to-µm scaling, conversion to 8-bit format, auto-thresholding using the MaxEntropy method [21], selection of thresholded particles (size filter = 200 pixels, to filter noise particles out), and particle analysis (count, area, perimeter, length, and width). Control versus CF comparison was achieved by an unpaired *t*-test using GraphPad Prism (version 7.00).

### 2.6. Experimental Design and Statistical Rationale

All sweat samples were collected under steady state conditions from subjects with no known acute or chronic illness in controls and no exacerbation in patients with CF, no drug (controls) or additional drug (CF) use at the time of collection, no cosmetic use or skin damage at the site of collection, no clinical sign of dehydration. Female subjects were neither pregnant nor lactating. Patients with CF had a confirmed diagnosis and were clinically stable, having a Forced Expiratory Volume in one second (FEV1) % predicted ≥ 30% and an O_2_ saturation ≥ 92%. Patients with CF, tested under stable conditions, were not enrolled in other clinical trials or under CFTR modulator therapies. All subjects were asked to be fasting and kept well-hydrated for a minimum of 8 h before collection.

After sweat collection, sweat chloride (coulometry, ChloroChek chloridometer, ELITechGroup, Brussels, Belgium), sodium and potassium concentrations (flame photometry, Flame Photometer Model 420, Sherwood Scientific, Cambridge, UK) were measured.

Due to the small and variable sweat volumes collected, together with the relatively low and variable protein concentrations of sweat, plus the need to store sweat samples for further analyses, no technical replicate could be performed for each sample. For the same reasons, no CF sample library was analyzed considering the high proteome similarity between the control and CF proteomes as well as for the use of the inter-sample match between the run option. To account for technical variability, sweat samples were processed and analyzed in three series of ten control samples plus a pooled sample and two series of seven and eight CF samples. The inter-series technical variability for a given group was negligible when compared to the biological variability [22]. Inter-sample technical variability was reduced by separating LFQ normalization between the control and CF groups: considering the equivalent amount of proteins processed and injected for all samples, the computation of both non-normalized data and separated LFQ normalized data drew the same general biological conclusions (both differential proteomic analysis and differential quantification of the quality control standard digest spike-in were also similar). However, global LFQ normalization led to a quantification bias (e.g., the 1:1:1:1 ratio of the standard protein digest spike-in was lost). Protein abundances between the two groups were not suitable for global normalization (Appendix A).

## 3. Results

### 3.1. The Proteome of CF Sweat

Sweat samples from 30 controls and 15 patients with CF—see Appendix A for complete clinical data summary—were analyzed by nanoLC-MS/MS (Figure 1). Based on chromatogram discrepancy and poor correlation with the other sample data, likely of technical origin, two female control samples and one male CF sample were discarded (Appendix A). Clinical data of the remaining control and patients with CF are summarized in Table 1 and Table 2. Considering a minimum of two peptides—including one unique peptide—and an FDR below 0.01 for protein identification, a total of 1057 proteins were identified, accounting for data filtering (Appendix A) and the standard protein mixture (MPDS Mix 1, Waters) for quality control. About 520 ± 18 (mean ± SEM, control: 542 ± 23, CF: 476 ± 26) proteins were peptide-spectrum matching hits while 317 ± 7 proteins (control: 310 ± 9, CF: 330 ± 11) required matching between runs for identification for an average total of 837 ± 14 proteins (control: 853 ± 18, CF: 805 ± 22) identified in each sample (Figure 2A). A total of 314 proteins were consistently identified across all samples.

The comparison of protein identifications between control and CF sweat proteomes emphasized a near-complete overlap and a high degree of similarity: 98% of proteins were common to the control and CF groups and only 18 out of 1057 identified proteins were exclusive to control sweat (Figure 2B). Exclusive proteins were in the low abundance and low occurrence tiers so one could not consider them biologically relevant. The classification and over-representation analysis of identified proteins highlighted the predominance of (i) proteins related to proteolytic activity, proteases, and peptidases as well as their respective inhibitors, (ii) cytoskeletal proteins, i.e., protein components and regulators (actin and Actin-Binding Proteins (ABP)) of the actin cytoskeleton organization and dynamics, (iii) proteins of reactive oxygen species metabolism and oxidative stress, (iv) markers of UPR and RE stress, (v) components and regulators of the proteasome, or (vi) proteins of all major metabolic pathways, among the over-represented proteins mapped to annotation clusters of the PANTHER Classification system and Gene Ontology Enrichment analysis, mapping protein IDs against PANTHER GO Slim annotation datasets (Appendix A).

### 3.2. Analysis of Sweat Proteome Profiles

A total of 1057 identified proteins were suitable for protein label-free quantification. For further statistical differential analysis between control and CF sample groups, only proteins identified and quantified in at least 50% of a sample group were used, amounting to 947 proteins.

First of all, the hierarchical clustering of control and CF samples together confirmed CF-specific/control-specific proteome profiles of PCC since samples were sorted into eight clusters, grouping samples from the same subject group (Figure 3A, upper panel), without a priori. Only four samples (one control and three CF) were mismatched. According to PERMANOVA, the variations in proteome profiles between control and CF were significantly correlated with Na^+^ and Cl^−^ concentrations, protein concentration, Na^+^ amount, and CF status (Figure 3A, lower panel). Then, the hierarchical clustering of CF samples into five clusters of CF proteome profiles highlighted the inter-individual biological variability of CF sweat (Figure 3B, upper panel). According to PERMANOVA, the variations in CF sweat proteome profiles were significantly correlated with sweat Na^+^ and K^+^ concentrations, protein concentration, and K^+^ amount (Figure 3B, lower panel). Interestingly, no significant correlation with CFTR genotype or clinical manifestations (pancreatic insufficiency, diabetes, airway infection status, or lung function impairment) was observed.

SPLS regression selected 135 out of 1057 proteins as important variables with high correlations with clinical data (Appendix A). Especially, SPLS confirmed the correlation between sweat ion composition and protein profiles (Appendix A).

### 3.3. Characterization of Candidate CF Biomarkers

The clustering of sweat proteome profiles like the sample grouping by PCA (Figure 4A) resulted from the differential abundance of 402 out of 947 proteins (Appendix A), as tested by a supervised two-sample *t*-test. CF was associated with a decrease in the expression of 351 out of 402 proteins in differential abundance, 51 out of 402 proteins being over-expressed. The proteome dynamic correlated with a partial depletion in CF sweat (as visualized in Figure 4B).

Of note, 17 of the 20 most abundant proteins in sweat [22] were in significantly differential abundance between the control and CF groups (Appendix A, highlighted in blue). In addition, kallikreins (KLK5, KLK11 (ESG-specific), KLK14) were significantly decreased in CF sweat. (Appendix A, highlighted in yellow).

One-hundred and eighty-nine proteins were differentially abundant between the control and CF groups (11 up in CF, 178 down, Appendix A) after a supervised two-sample *t*-test with permutation-based FDR calculation. Data filtering based on sample occurrence (protein found in all samples) significance (*p* < 0.001) and value (control-CF log_2_ LFQ difference ≥ 2 or control-CF log_2_ LFQ difference ≤ −2) of the difference in protein abundance highlighted nine candidate CF biomarkers of potential clinical relevance (Figure 4). Decreases in the protein abundances of ARG1, CUTA, MAN1A1, AGA, EZR, SIAE, LFNG, MIA3, and FNLA (Figure 4C, panel a) between the control and CF groups were the most statistically significant. Only EZR and FLNA were closely functionally related, being involved in the integrity and stability of the cortical actin network.

According to the PANTHER-Gene Ontology functional annotation and over-representation test (Appendix A), proteins involved in: (i) the hydrolase activity (both protease/peptidase and glycosidase activity) of the lysosome, (ii) the structure and function of the proteasome, (iii) protein processing and mechanisms of ER stress and UPR, (iv) the structure of desmosomal anchoring junctions, and (v) the structure, organization, and dynamics of the actin cytoskeleton were over-represented in the set of differentially abundant proteins (Figure 5).

Of note, the differential phenotypes of CFTR/F508del CFTR physical and functional interactions could be described in sweat since subsets of sweat CF biomarkers cross-checked the CFTR interactome (*n* = 82, Figure 6A) and F508del CFTR interactome (*n* = 23, Figure 6B) as established by Pankow et al. [23].

#### CF Biomarker Profiles Partially Reflect CF Severity Related to CFTR Genotype

To determine if CF biomarker profiles correlate with genotype, pancreatic status, airway infection status, or spirometry, computation of Pearson’s correlation coefficients, average Euclidian distance hierarchical clustering of samples, and PERMANOVA testing of available clinical parameters were applied to CF biomarkers only (*n* = 402).

Firstly, the hierarchical clustering of all samples using only the protein abundance profiles of CF biomarkers generated CF and control groups sub-divided into five clusters of protein abundance profiles (Appendix A, upper panel), without a priori. According to PERMANOVA, the variations in CF biomarker profiles between control and CF were significantly correlated with sweat Na^+^, Cl^−^, and K^+^ concentrations, protein concentration, Na^+^, Cl^−^, and K^+^ amounts, and CF status (Appendix A, lower panel). Secondly, the hierarchical clustering of CF samples (*n* = 14) using only the protein abundance profiles of CF biomarker sorted samples into six clusters (Appendix A, upper panel), without a priori. According to PERMANOVA, the variations in CF sweat proteome profiles were significantly correlated with CFTR genotype (F508del homozygous status) (Appendix A, lower panel). No significant correlation with clinical manifestations (pancreatic insufficiency, diabetes, airway infection status, or lung function impairment) was observed.

### 3.4. Characterization of CF Severity Biomarkers

The correlation between sweat CF biomarker profiles and CFTR genotype, as it was observed by PERMANOVA of hierarchical clustering, resulted from the differential abundance of 68 proteins between F508del homozygous patients and patients with other genotypes (Appendix A), as tested by a supervised two-sample *t*-test. Dermcidin (DCD), the most abundant protein in sweat, was significantly less abundant in F508del homozygous sweat (Appendix A, highlighted in blue).

Concurrently, applying the same supervised statistics to patients with or without pancreatic insufficiency, 71 proteins were found differentially abundant (Appendix A).

According to the PANTHER-Gene Ontology functional annotation and over-representation test, proteins involved in: (i) the hydrolase activity (both protease/peptidase and glycosidase activity) of the lysosome and (ii) the structure and function of the ribosome were over-represented in both F508del homozygous and PI patients (Appendix A).

Meanwhile, 54 proteins were differentially abundant in correlation with lung function, between patients with normal or mildly impaired lung function (≥70% FEV_1_%) and patients with moderate to severe lung function impairment (<70% FEV_1_%) (Appendix A). According to the PANTHER-Gene Ontology functional annotation and over-representation test, hydrolases and desmosomal linkers to intermediate filaments were over-represented in this subset (Appendix A).

Data filtering based on sample occurrence (protein found in all samples) significance (*p* < 0.001) and value (control-CF log_2_ LFQ difference ≥ 2 or control-CF log_2_ LFQ difference ≤ −2) of the difference in protein abundance highlighted seven candidate biomarkers of CF severity with potential clinical relevance (Figure 4). Differential abundances of: (i) ARG1, GPT, MDH2, and EML4 between F508del homozygous and F508del heterozygous patients, (Figure 4C, panel b), (ii) MGAT1 between pancreatic insufficient and pancreatic sufficient patients (Figure 4C, panel c), and (iii) IGJ and TOLLIP between patients with normal lung function/mild lung function impairment and moderate/severe lung function impairment (Figure 4C, panel d) were the most statistically significant. 

Nevertheless, the latter results were generated from a small number of patients and would gain clinical relevance with a larger cohort and subsequent dataset.

### 3.5. Actin Dynamics in CF Sweat

In light of the functional annotation of the sweat proteome, 35 out of all identified proteins are involved in the organization and dynamics of the actin cytoskeleton (Appendix A). Supervised differential proteomics reported 13 ABPs with significant changes in protein abundance between the control and CF groups while actin abundance remained steady between healthy subjects and patients with CF. 

ABPs cofilin-1 (CFL), insulin receptor substrate 53 kDa/brain-specific angiogenesis inhibitor 1-associated protein 2 (IRSp53/BAIAP2), and dihydropyrimidinase-related protein 3 (DPSYL3) plus actin-bundling protein lysozyme C (LYZ) were more abundant in CF sweat.

ABPs tropomyosins 1 and 3, plastins 2 and 3 (LCP1 and PLS3), plus small RhoGTPase Ras-related C3 botulinum toxin substrate 1 (RAC1), and myotrophin (MTPN) were less abundant in CF sweat. 

At the same time, subunits of the Arp2/3 complex were differentially abundant from one another: Arp2/3 complex subunit 1A (ARPC1A) was less abundant in CF sweat while actin-related protein 2 (ACTR 2), Arp2/3 complex subunit 3 (ARPC3), and Arp2/3 complex subunit 4 (ARPC4) were more abundant. Actin-related protein 3 (ACTR3) and Arp2/3 complex subunit 2 (ARPC2) abundances were not significantly different between the control and CF groups.

In regard to the differential abundance of ABPs and functionally associated proteins, the observation of sweat F-actin (Figure 7A) featured significant differences in microfilament organization between control and CF sweat. In detail, particles assimilated to microfilaments in CF sweat were significantly more abundant (Figure 7B, panel a, fold change (FC) = 1.81, *p* < 0.001 ***), longer and larger with significantly greater mean microfilament area (Figure 7B, panel b, FC = 2;68, *p* < 0.001 ***), perimeter (Figure 7B, panel c, FC = 1.92, *p* < 0.001 ***), length (Figure 7B, panel d, FC = 1.79, *p* < 0.001 ***), and width (Figure 7B, panel e, FC = 1.69, *p* < 0.001 ***) and significantly lower circularity (Figure 7B, panel f, FC = 0.80, *p* < 0.001 ***, circularity = 1 describes a perfectly round object).

## 4. Discussion

The current study was designed to achieve the first thorough and in-depth characterization of the sweat proteome of patients with CF versus healthy subjects, at an individual level. A standardized and optimized workflow from the sample collection and preparation to the LC-MS analysis and bio-informatic data processing was developed. With a particular emphasis on sweat sampling in a non-invasive and reproducible way, the methodology followed the Gibson and Cooke sweat test gold standard guidelines. Accounting for all analyzed samples, 1057 proteins were identified and quantified by their inter-individual relative abundance. Comparing 14 patients with CF to 28 healthy subjects as a control cohort, relative protein abundances between control and CF sweat samples were computed to: (i) evaluate whether sweat protein profiles would help discriminate patients with CF from healthy subjects, (ii) test the correlation between CF severity and sweat protein profiles, and (iii) characterize biomarkers of CF disease and severity in sweat.

First and foremost, the statistical analysis of sweat proteome profiles without a priori partially reflected the clinical diagnosis and conclusions of the sweat test since CF status alongside Na^+^ and Cl^−^ concentrations correlated with the control versus CF sample clustering. However, the clinical relevance of sweat proteome profiles was hindered by the respective inter-individual variabilities of control sweat—inter-subject variability correlated with Na^+^ amount [22]—and CF sweat—inter-patient variability correlated with K^+^ concentration and K^+^ amount alongside Na^+^ concentration and sweat protein concentration. On a side note, these correlations underlined the susceptibility of protein content to the shifts in the mechanisms of eccrine ion secretion between control subjects and patients with CF. The fact remains that the combined physiological and pathophysiological inter-individual variability of sweat allowed control versus CF sample correlation and clustering mismatches, with no evidence to rationalize or exclude mismatched samples as outliers. Therefore, the statistical analysis of whole sweat proteome profiles without a priori could not discriminate patients with CF from the healthy population. Neither could it discriminate against patients based on their genotype or the clinical manifestations, failing to report CF severity.

However, CF samples were distributed together without a priori among clusters in correlation with CF status, even if a priori control and CF sample groups were not retained afterward. So, CF pathophysiology partly influenced the protein composition of sweat. Applying a supervised statistical analysis for differential proteomics between the CF and control groups, a third of all characterized sweat proteins (351 out of 947) were significantly less abundant in CF sweat. Functionally wise, the over-representation of differentially abundant proteins involved in protein processing, ER stress, and UPR pathways was consistent with this apparent depletion. This observation had to be correlated with phenotypes of impaired CFTR processing due to *CFTR* mutations and generalized to the global protein machinery of eccrine gland cells. Thirteen out of fourteen patients with CF carried the F508del mutation whose phenotype is the absence of functional CFTR at the membrane due to CFTR production and trafficking impairment and degradation of misfolded proteins. The over-representation of proteins in disease-specific abundance related to proteasome- and lysosome-mediated proteolysis alongside markers of UPR was in total agreement with the prevalence of the F508del mutation in the patient cohort.

To sum up, the protein composition of CF sweat compared to control sweat highlighted that CF pathophysiology of the eccrine gland, e.g., defects in the mechanisms of protein processing resulting in ER stress, the onset of the UPR, and proteolysis can be indirectly monitored by sweat proteomics. From a pathophysiological standpoint, the F508del mutation globally affected the protein machinery of the eccrine gland beyond the sole processing of CFTR, hence the disease-driven depletion of the CF sweat proteome.

When considering the 402 differentially abundant proteins, the proteome profiles of CF sweat correlated with *CFTR* F508del mutation, i.e., sweat was a matrix of candidate biomarkers of both CF diagnosis and discrimination between F508del homozygous and F508del heterozygous status. The pathophysiology of F508del homozygous patients is frequently associated with the onset of pancreatic insufficiency [24]. Interestingly, the most differentially abundant proteins were both F508del homozygous and PI biomarkers. Yet, the recruitment of F508del heterozygous patients with PI helped characterize F508del homozygous- and PI-specific biomarkers. The over-representation of proteins in genotype- or PI- related abundance involved in the protease activity of the cytoplasm and the structure and function of the ribosome echoed the correlation between CFTR mutation classes, phenotypes of protein processing defects, and CF severity.

In brief, the protein composition of CF sweat highlighted that factors of CF severity (CFTR genotype) can be monitored by sweat proteomics. From a pathophysiological perspective, ribosomal stalk proteins were described as modifiers of CF severity when the silencing of corresponding genes elicited the partial phenotype rescue of F508del CFTR processing defects [25]. Here, ribosomal stalk proteins uL11, P0 (uL10), and P2 plus ribosomal proteins uL4 and eL6 were sweat markers of CF disease and severity, respectively.

As for the pathophysiological relevance of sweat proteins in differential abundance, 17 proteins from the core proteome (systematically found in all samples) and the top 20 most abundant proteins of sweat [22] were in CF-specific abundance. To a greater extent, proteins in the high abundance and high occurrence tiers were all affected by CF. Dermcidin, the most abundant protein in sweat was characterized as a biomarker of F508del homozygous and PI. Moreover, a number of previous results about CF-specific abundance of proteins were confirmed: (i) the decreased levels of kallikreins were already described in sweat [26] and correlated with a decreased enzymatic activity in CF plasma [27], (ii) the decreased arginase-1 abundance in CF sweat would correlate with the decreased abundance and reduced enzymatic activity in CF plasma [28], (iii) the under-expression of filamins A and B in CF sweat could be correlated with the decreased cell levels of filamins [29]. Interestingly, in the current work, arginase-1 and filamin A were characterized as promising candidate biomarkers for CF diagnosis and estimation of CF severity from CF sweat. More importantly, some biomarkers found in CF sweat were already described in previous studies on the secretome and proteome of bronchial epithelial cell lines [30,31,32], the proteome of nasal epithelial cells [33], the proteome of CF serum [34], and the proteome of CF urine exosomes [35] (Appendix A). Inconsistencies in the protein abundances of these biomarkers were observed between studies and could be partially explained by the nature of the models (e.g., in vitro cell lines versus patient tissue samples). Still, proteins related to CFTR proteostasis and membrane stability plus cytoskeleton crosstalk (e.g., FLNA, EZR, VCL, SET, COL6A1, and HSPA5) were among the ones in good agreement throughout. In total, 82 sweat proteins in CF-specific abundance (Appendix A) were listed in the CFTR interactome [23]. In addition, in CF sweat, the decreased levels of CFTR interactors, e.g., ERM proteins (EZR, MSN) or filamins, plus the under-expression of all the protein constituents of the desmosome highlighted: (i) the defects in CFTR homeostasis and membrane stability as well as cell-cell adherence and membrane integrity of the eccrine gland cells, (ii) the potential of sweat analysis to remotely monitor some aspects of CF pathophysiology in other epithelia. In addition, from a clinical standpoint, sweat CFTR interactors and F508del CFTR interactors (Appendix A) in CF-specific abundance are of utmost interest in the search for biomarkers of phenotypic rescue to benefit new therapeutic developments.

Concurrently, the functional annotation of proteins in disease-specific abundance pointed out new insights into the pathophysiology of CF sweat, i.e., the significant over-representation of proteins related to sweat actin organization and dynamics.

Precisely, protein abundances of lysozyme C (LYZ) and Actin-Binding Protein (ABP) cofilin-1 (CFL) were significantly increased in CF sweat while actin levels remained steady. Both LYZ and CFL are known to alter actin dynamics in relation to variations in ionic strength and concentration ratio to actin, as discussed below and summarized in Figure 8. Changes in LYZ and CFL abundance correlated with CF-specific changes in sweat actin organization, i.e., larger, more abundant microfilaments resulting from disease-induced F-actin polymerization and bundling.

Initially described as an Actin-Depolymerizing Factor (ADF) [36], CFL regulates actin dynamics as a whole from F-actin/G-actin balance and treadmilling to the spatial organization of the actin cytoskeleton [37,38]. CFL concentration ratio to actin (CFL: ACT ratio), pH, oxidative stress, or CFL phosphorylation are well-described modulators of CFL actin-related biological function and activity [39]. 

At a higher CFL: ACT ratio, F-actin microfilaments are severed and prone to the nucleation and polymerization of new branches. Conversely, F-actin microfilaments are severed and completely depolymerized into G-actin at a lower CFL: ACT ratio [40,41]. In CF sweat, the higher CFL abundance in relation to steady actin levels increased CFL:ACT ratio. So, CF sweat G-actin was more likely to nucleate and be polymerized into F-actin while microfilaments were more likely to elongate.

This phenomenon would be accentuated by CFL sensitivity to pH [42]. At a slightly acidic pH (6.5), CFL is less likely to divert actin dynamics from nucleation and polymerization [40]. Since sweat pH is neutral to slightly acidic varying between pH 5 (at the lowest sweat rate) and pH 7 [43], with no significant change correlated with CF [44], F-actin microfilaments would be more likely to stabilize and elongate further.

In the meantime, tropomyosins, known competitors of CFL for access to actin microfilaments [45,46], and MTPN, an inhibitor of actin polymerization via interaction with actin-capping proteins [47], were decreased in CF sweat.

Although (de)phosphorylation of its Ser3 residue is an important modulator of intracellular CFL activity, it is noteworthy that no protein involved in CFL regulation by (de)phosphorylation (slingshot phosphatases and LIM kinases) was identified in control or CF sweat. Interestingly enough, an upstream regulator of LIM kinases, small RhoGTPase Rac1, was under-expressed in CF sweat. On that note, Rac1 is involved in the regulation of both actin polymerization by CFL and actin bundling by IRSp53/BAIAP2. IRSp53 bundles F-actin microfilaments by means of its SH3 domain (recruitment of other ABPs and actin regulators) and its IMD domain (direct actin binding) [48,49]. The respective CF-related abundances of Rac1 (down) and IRSp53 (up) suggested a promotion of IMD-mediated actin bundling by IRSp53 without regulation.

In addition to other actin-bundling proteins, DPYSL3 [50], LCP1, and PLS3 in particular exhibited CF-associated abundance, suggesting that increased actin bundling resulted from IRSp53, DPYSL3, and LYZ-bound microfilaments.

Indeed, polycationic proteins and peptides such as LYZ and cathelicidin-derived LL-37 were described as promoters of F-actin nucleation, polymerization, and bundling [51,52]. F-actin microfilaments are bound together through electrostatic (hydrophobic) interactions, mediated by intercalated polycations, with sensitivity to changes in ionic strength, polycation concentration, and actin concentration [53]. Polycation-induced F-actin bundling is promoted by an increase in polycation concentration while prevented by an increase in ionic strength or actin concentration. Sweat ionic strength is mainly influenced by NaCl concentration. In physiological conditions, sweat is hypotonic due to ion reabsorption along the duct of the eccrine gland. In CF, sweat is hypertonic (up to plasma concentration) as the absence of a functional CFTR prevents Cl reabsorption. The presence of polycation-bound F-actin bundles has already been described in the pathologically viscous sputum [54]—a bio-fluid with similar neutral to slightly acidic pH [55,56] and increased ionic strength [57] as CF sweat-secreted and accumulated in the pulmonary airways of patients with CF. In the same way, reports of the higher viscosity of CF sweat [44] could be explained by extensive LYZ-mediated F-actin bundling of polymerizing, unbranched microfilaments.

Besides their interaction with actin, polycationic proteins and peptides are also known for their antimicrobial properties [58], essential to skin host defenses and the innate immune system. While trapped in F-actin bundles, polycationic proteins and peptides reversibly lose their antimicrobial activity [59] and potentiate the risk of pulmonary infection linked to higher mucus viscosity in patients with CF. Nonetheless, the prevalence of skin infections in patients with CF does not corroborate defects in innate immune defenses to take place as in CF sputum and airways. The predominance of dermcidin (DCD)—a polyanionic precursor of antimicrobial peptides and the most abundant protein in sweat—in skin defenses together with the absence of CF changes in DCD abundance could explain these observations [60,61,62].

Interestingly, in the event of a deep correlation, yet to be established, between sweat and mucus F-actin bundling states and subsequent viscosities, the monitoring of sweat F-actin dynamics could prove useful in the early evaluation of CF severity.

To sum up, CF was correlated with disease-specific proteome profiles, contributing to non-physiological inter-individual variations of the sweat proteome. The characterization of differentially expressed proteins (control vs. CF, F508del heterozygous vs. F508del homozygous, PS vs. PI, normal/mild vs. moderate/severe lung function impairment) showcased nine CF diagnosis biomarkers (CUTA, ARG1, EZR, AGA, FLNA, MAN1A1, MIA3, LFNG, SIAE) and seven CF severity biomarkers (ARG1, GPT, MDH2, EML4 (F508del homozygous), MGAT1 (pancreatic insufficiency), TOLLIP, IGJ (lung function impairment)) candidates as well as potential markers of CFTR phenotypic rescue to be further investigated for clinical relevance. On that note, particular attention to the pathophysiology of ABPs in sweat compared to other CF tissue would also deserve further investigations. In conclusion, sweat proved to be an informative bio-fluid to help and improve the understanding of CF pathophysiology by providing candidate biomarkers of interest for precision medicine and therapeutic development.

## Figures and Tables

**Figure 1 cells-11-02358-f001:**
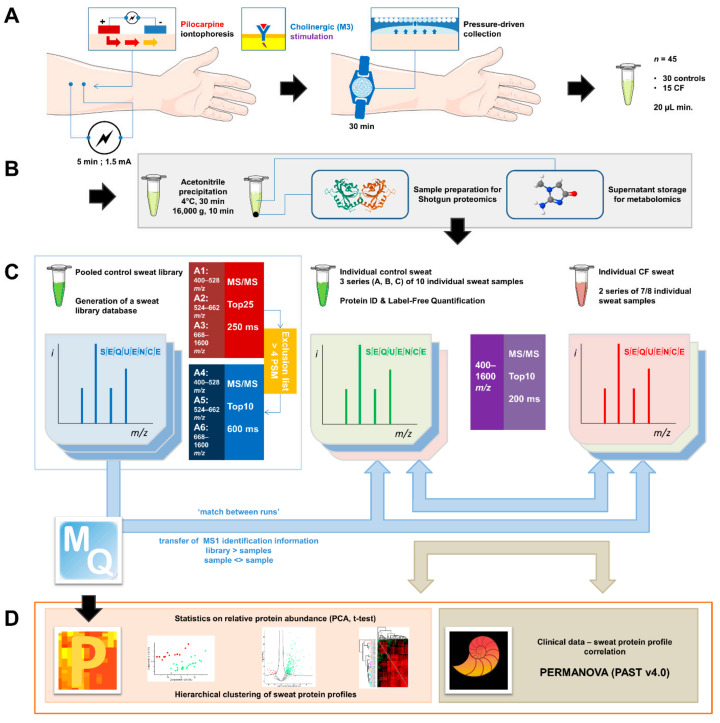
General experimental workflow. (**A**) Standardized sweat collection method. (**B**) Single sample preparation method for subsequent sweat proteomics and metabolomics studies. (**C**) Analytical and bioinformatic strategy using the “match between runs” option (MaxQuant). 1. Generation of a sweat reference proteome database from pooled control samples. A two-round analysis of three limited *m*/*z* range acquisitions was performed. A precursor exclusion list was applied for the second-round experiment. 2. Individual sweat sample analyses. (**D**) Statistical data processing tools.

**Figure 2 cells-11-02358-f002:**
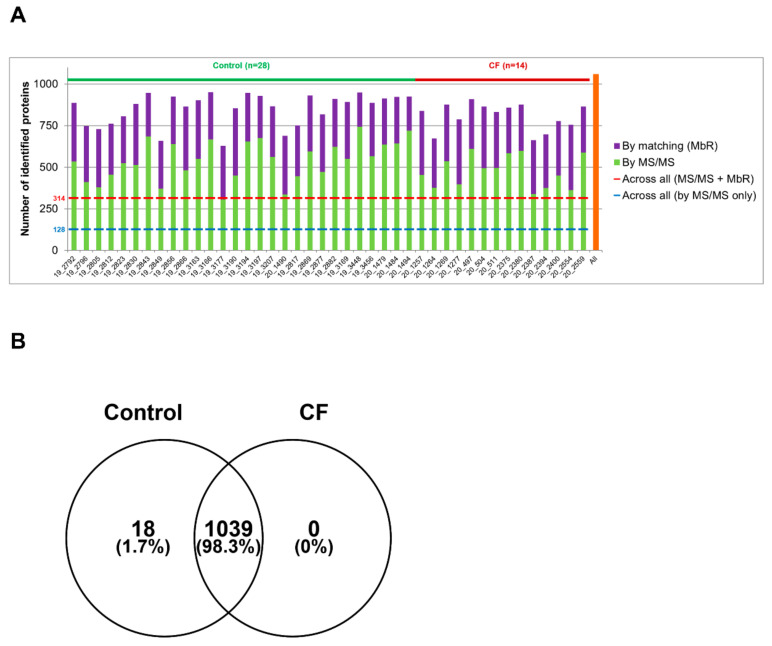
High similarity of control and CF proteome in protein identification (**A**). Number of proteins identified in each sample by MS/MS (in green) and by MbR (in purple), with the total number of identified proteins (in orange). (**B**) Protein identification overlap between control and CF samples.

**Figure 3 cells-11-02358-f003:**
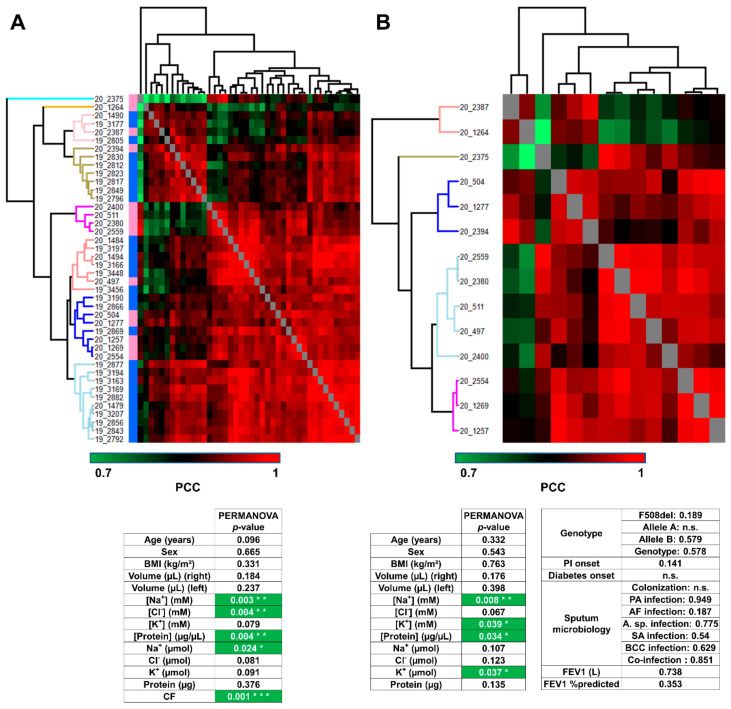
Sweat protein profiles discriminated patients with CF from control subjects, without correlation with CF severity and complications (**A**). Heat-map representation of control versus CF sweat protein profiles. Control samples (blue color bar), CF samples (pink color bar). (**B**) Heat-map representation of CF sweat protein profiles. Hierarchical clustering of Pearson’s correlation coefficients using average Euclidian distance matrix. PERMANOVA test for significance of the correlation between clustering and clinical data distribution (number of permutations = 999, *p* < 0.05 *, *p* > 0.01 **, *p* > 0.001 ***), ion formulae into brackets indicate ion concentrations and ion formulae without brackets indicate ion amounts.

**Figure 4 cells-11-02358-f004:**
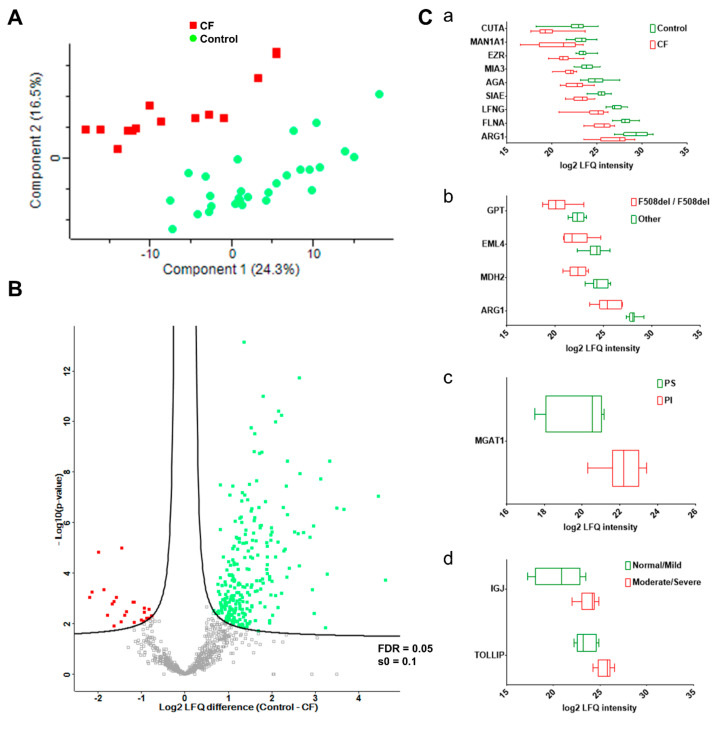
CF status and severity correlated with disease-specific abundances of sweat proteins. (**A**) PCA, CF (plain red squares), control (plain green dots). (**B**) Volcano plot, CF-overexpressed (plain red squares), CF-underexpressed (plain green squares), threshold curve (black bold line, FDR = 0.05, s0 = 0.1). (**C**) Box-and-whiskers representation of sweat protein abundances for (**a**) CF, (**b**) F508del homozygous, (**c**) PI, and (**d**) lung function candidate biomarkers.

**Figure 5 cells-11-02358-f005:**
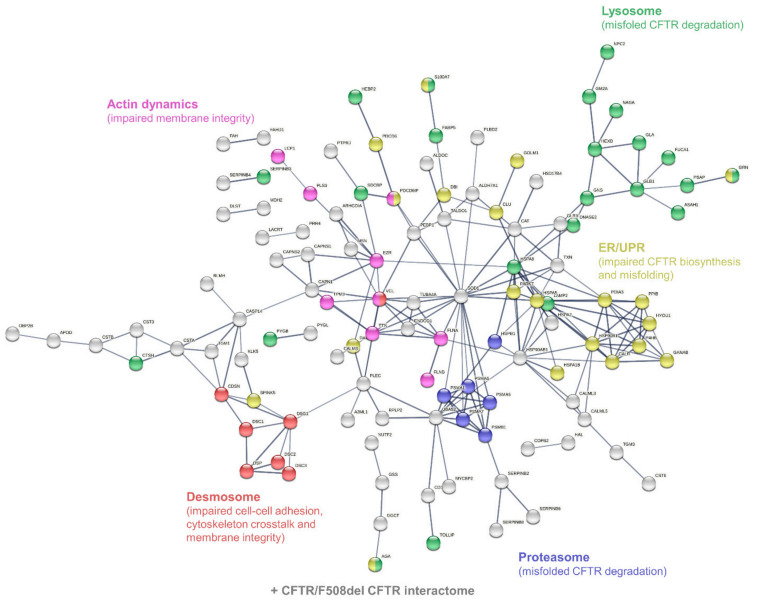
Interaction network mapping of sweat core CF biomarkers (182 proteins in 42/42 samples). The 182 query proteins resulted in 180 mapped proteins. Network settings included: Homo sapiens database, high confidence minimum required interaction score, hidden disconnected nodes, and confidence-based networking.

**Figure 6 cells-11-02358-f006:**
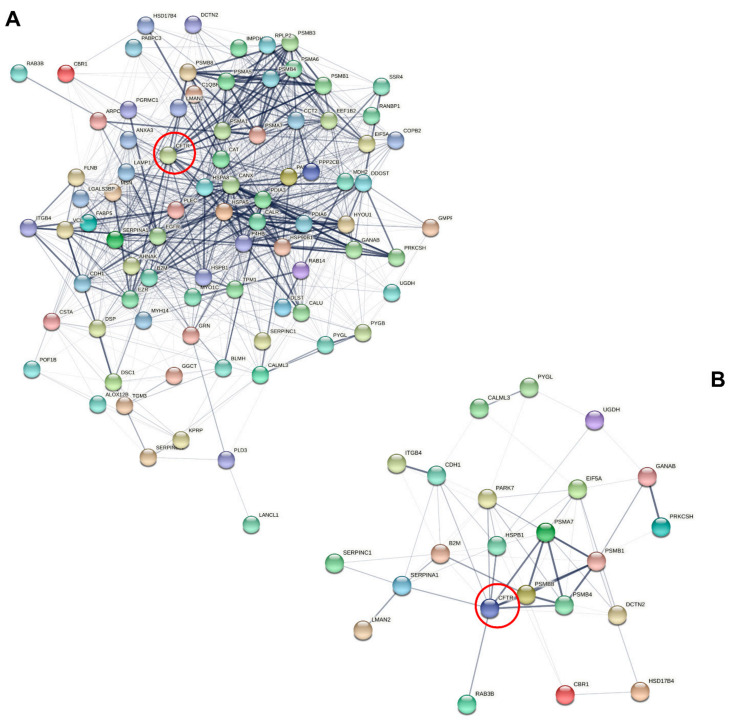
Interaction network mapping of the CFTR interactome and F508del CFTR interactome of sweat CF biomarkers. (**A**) Sweat CFTR interactors in CF-specific abundance (82/82 mapped proteins). (**B**) Sweat F508del CFTR interactors in CF-specific abundance (22/23 mapped proteins). Network settings included: Homo sapiens database, lowest confidence minimum required interaction score, hidden disconnected nodes, and confidence-based networking. CFTR was added to the list of sweat proteins of interest before analysis (red circle).

**Figure 7 cells-11-02358-f007:**
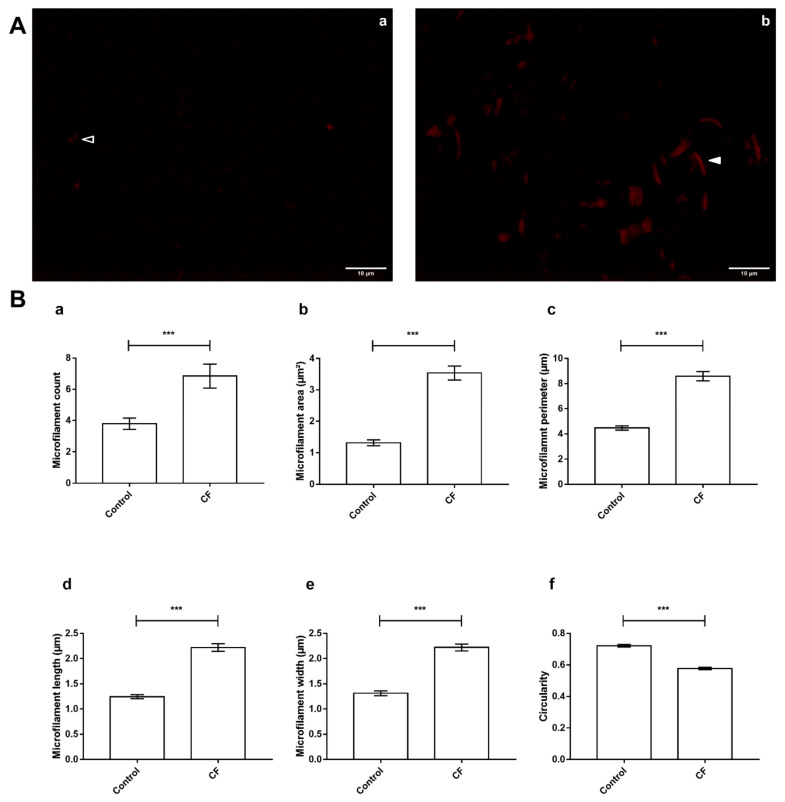
CF-specific abundance of sweat ABP equated to differences in the organization of F-actin microfilaments between control and CF sweat. (**A**) Specimen epifluorescence micrographs of free-in-solution Phalloidin-TRITC-labeled F-actin microfilaments in control (**a**) and CF (**b**) sweat. Arrowheads: control (empty) and CF (plain) microfilament specimens, scale bar = 10 µm. (**B**) Mean count (**a**), area and perimeter (**b**,**c**), length (**d**), width (**e**), and circularity (**f**) of thresholded particles assimilated to F-Actin microfilaments, between control and CF sweat. Mean ± S.E.M. Student’s *t*-test for statistical significance analysis: *** *p* < 0.001, (*n* = 5, 5 controls compared to 5 patients with CF; n_micrographs_ = 20 per individual).

**Figure 8 cells-11-02358-f008:**
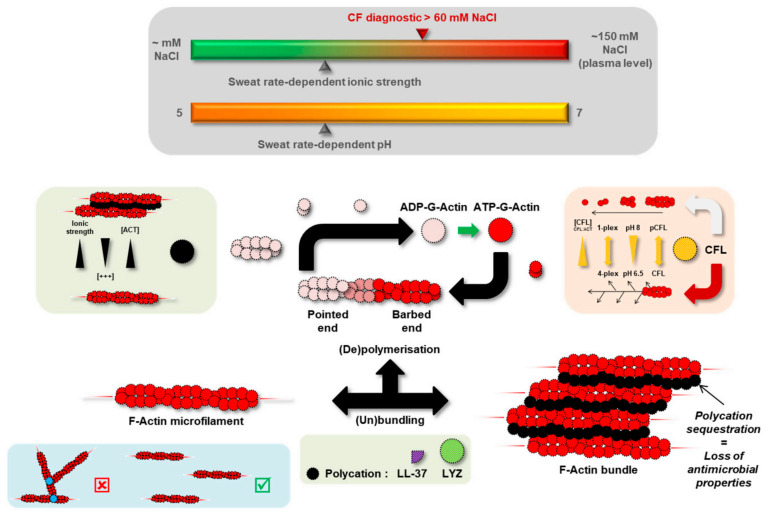
Hypothetical mechanism of F-actin microfilament reorganization in CF sweat.

**Table 1 cells-11-02358-t001:** Clinical data summary. Student’s *t*-test, significantly different parameters highlighted in green, *p* < 0.05 *, *p* < 0.01 **, *p* < 0.001 ***. Green (control) and red (CF) shades of background color indicated the different analytical series.

Sample IDs	Age (Years)	Sex	BMI (kg/m^2^)	Collected Volume (µL) (Right Arm)	Collected Volume (µL) (Left Arm)	[Na^+^] (mM)	[Cl^−^] (mM)	[K^+^] (mM)	[Protein] (µg/µL)	Na^+^ (µmol)	Cl^−^ (µmol)	K^+^ (µmol)	Protein (µg)	
**19_2792**	**36**	**M**	**22.7**	**62.1**	**87.8**	**56**	**26**	**8**	**0.488**	**3.48**	**1.61**	**0.50**	**42.85**	**Control** **(*n* = 28)**
**19_2796**	**29**	**M**	**31.6**	**48.1**	**42.4**	**23**	**4**	**10**	**0.557**	**1.11**	**0.19**	**0.48**	**23.62**	
**19_2805**	**28**	**M**	**30.0**	**98.9**	**97.5**	**59**	**24**	**7**	**0.340**	**5.84**	**2.37**	**0.69**	**33.15**	
**19_2812**	**40**	**M**	**25.5**	**89.6**	**96.5**	**39**	**18**	**8**	**0.432**	**3.49**	**1.61**	**0.72**	**41.69**	
**19_2866**	**74**	**M**	**21.3**	**56.9**	**64.2**	**37**	**12**	**12**	**0.551**	**2.11**	**0.68**	**0.68**	**35.37**	
**19_2823**	**31**	**F**	**19.8**	**71.8**	**89.8**	**32**	**10**	**10**	**0.262**	**2.30**	**0.72**	**0.72**	**23.53**	
**19_2830**	**30**	**F**	**23.1**	**94.0**	**77.9**	**32**	**11**	**8**	**0.462**	**3.01**	**1.03**	**0.75**	**35.99**	
**19_2843**	**24**	**F**	**18.2**	**63.5**	**64.1**	**65**	**36**	**10**	**0.472**	**4.13**	**2.29**	**0.64**	**30.26**	
**19_2849**	**27**	**F**	**20.5**	**97.7**	**84.6**	**22**	**10**	**8**	**0.432**	**2.15**	**0.98**	**0.78**	**36.55**	
**19_2856**	**29**	**F**	**18.7**	**79.7**	**70.4**	**28**	**6**	**8**	**0.422**	**2.23**	**0.48**	**0.64**	**29.71**	
**19_3163**	**32**	**F**	**19.4**	**44.8**	**30.1**	**30**	**12**	**9**	**0.785**	**1.34**	**0.54**	**0.40**	**23.63**	
**19_3166**	**26**	**F**	**18.9**	**57.2**	**64.0**	**44**	**22**	**10**	**0.494**	**2.52**	**1.26**	**0.57**	**31.62**	
**20_1490**	**39**	**F**	**21.3**	**40.1**	**43.5**	**26**	**12**	**8**	**0.371**	**1.04**	**0.48**	**0.32**	**16.14**	
**19_3177**	**28**	**M**	**23.1**	**35.9**	**27.0**	**51**	**30**	**8**	**0.475**	**1.83**	**1.08**	**0.29**	**12.83**	
**19_3190**	**29**	**M**	**24.5**	**98.6**	**80.7**	**55**	**28**	**6**	**0.406**	**5.42**	**2.76**	**0.59**	**32.76**	
**19_3194**	**41**	**M**	**24.1**	**34.4**	**42.7**	**36**	**8**	**9**	**0.736**	**1.24**	**0.28**	**0.31**	**31.43**	
**19_3197**	**36**	**M**	**22.5**	**27.5**	**82.7**	**91**	**44**	**8**	**0.504**	**2.50**	**1.21**	**0.22**	**41.68**	
**19_3207**	**28**	**M**	**23.8**	**84.9**	**93.2**	**45**	**22**	**7**	**0.364**	**3.82**	**1.87**	**0.59**	**33.92**	
**20_1494**	**29**	**F**	**20.9**	**61.3**	**64.4**	**54**	**30**	**12**	**0.441**	**3.31**	**1.84**	**0.74**	**28.40**	
**19_2869**	**28**	**F**	**17.7**	**57.1**	**45.9**	**67**	**10**	**24**	**0.646**	**3.83**	**0.57**	**1.37**	**29.65**	
**19_2877**	**33**	**F**	**22.3**	**102.7**	**83.7**	**35**	**12**	**7**	**0.350**	**3.59**	**1.23**	**0.72**	**29.30**	
**19_2882**	**24**	**F**	**20.6**	**57.2**	**53.6**	**29**	**10**	**6**	**0.377**	**1.66**	**0.57**	**0.34**	**20.21**	
**19_3169**	**57**	**F**	**19.2**	**35.6**	**29.9**	**58**	**24**	**8**	**0.661**	**2.06**	**0.85**	**0.28**	**19.76**	
**20_1479**	**25**	**M**	**24.5**	**56.4**	**72.6**	**69**	**38**	**7**	**0.336**	**3.89**	**2.14**	**0.39**	**24.39**	
**20_1484**	**24**	**M**	**23.6**	**99.0**	**93.4**	**49**	**20**	**8**	**0.622**	**4.85**	**1.98**	**0.79**	**58.09**	
**19_2817**	**26**	**M**	**20.8**	**47.4**	**40.4**	**73**	**44**	**9**	**0.314**	**3.46**	**2.09**	**0.43**	**12.69**	
**19_3448**	**30**	**M**	**20.6**	**91.2**	**74.0**	**50**	**22**	**10**	**0.448**	**4.56**	**2.01**	**0.91**	**33.15**	
**19_3456**	**31**	**M**	**22.6**	**101.5**	**99.5**	**69**	**40**	**5**	**0.198**	**7.00**	**4.06**	**0.51**	**19.70**	
**20_497**	**36**	**M**	**24.7**	**98.5**	**92.1**	**122**	**106**	**14**	**0.492**	**12.02**	**10.44**	**1.38**	**45.31**	**CF** **(*n* = 14)**
**20_504**	**32**	**M**	**26.8**	**74.6**	**85.8**	**106**	**100**	**17**	**0.424**	**7.91**	**7.46**	**1.27**	**36.38**	
**20_511**	**57**	**M**	**25.3**	**72.0**	**34.6**	**75**	**56**	**9**	**0.472**	**5.40**	**4.03**	**0.65**	**16.33**	
**20_1257**	**43**	**M**	**25.4**	**55.3**	**71.4**	**130**	**102**	**9**	**0.208**	**7.19**	**5.64**	**0.50**	**14.85**	
**20_1264**	**31**	**M**	**20.9**	**54.0**	**30.2**	**103**	**92**	**14**	**1.315**	**5.56**	**4.97**	**0.76**	**39.71**	
**20_1269**	**55**	**M**	**20.5**	**59.9**	**53.1**	**151**	**114**	**8**	**0.237**	**9.04**	**6.83**	**0.48**	**12.58**	
**20_1277**	**23**	**F**	**20.8**	**103.4**	**102.7**	**106**	**102**	**13**	**0.414**	**10.96**	**10.55**	**1.34**	**42.52**	
**20_2375**	**46**	**F**	**21.6**	**14.1**	**43.2**	**67**	**61**	**13**	**1.060**	**0.94**	**0.86**	**0.18**	**45.79**	
**20_2380**	**35**	**M**	**18.5**	**62.9**	**90.3**	**109**	**84**	**14**	**0.489**	**6.86**	**5.28**	**0.88**	**44.16**	
**20_2387**	**47**	**F**	**27.9**	**64.8**	**71.3**	**72**	**68**	**10**	**0.512**	**4.67**	**4.41**	**0.65**	**36.51**	
**20_2394**	**30**	**M**	**25.0**	**77.4**	**75.2**	**110**	**100**	**16**	**0.515**	**8.51**	**7.74**	**1.24**	**38.73**	
**20_2400**	**63**	**F**	**19.7**	**52.2**	**55.3**	**98**	**82**	**15**	**0.404**	**5.12**	**4.28**	**0.78**	**22.34**	
**20_2554**	**21**	**M**	**23.7**	**89.9**	**103.4**	**142**	**108**	**9**	**0.287**	**12.77**	**9.71**	**0.81**	**29.68**	
**20_2559**	**54**	**M**	**19.8**	**18.6**	**47.6**	**123**	**94**	**15**	**0.504**	**2.29**	**1.75**	**0.28**	**23.99**	
**Mean**	**33**	**13 F** **15 M**	**22.2**	**67.7**	**67.7**	**47**	**21**	**9**	**0.462**	**3.13**	**1.39**	**0.58**	**29.72**	
**Median**	**29**		**21.8**	**61.7**	**71.5**	**47**	**21**	**8**	**0.445**	**3.16**	**1.22**	**0.59**	**29.98**	
**SD**	**11**		**3.2**	**24.2**	**22.7**	**18**	**12**	**3**	**0.137**	**1.49**	**0.89**	**0.24**	**9.87**	
**SEM**	**2**		**0.6**	**4.6**	**4.3**	**3**	**2**	**1**	**0.026**	**0.28**	**0.17**	**0.05**	**1.87**	
**Mean**	**41**	**4 F** **10 M**	**22.9**	**64.1**	**68.3**	**108**	**91**	**13**	**0.524**	**7.09**	**6.00**	**0.80**	**32.06**	
**Median**	**40**		**22.7**	**63.9**	**71.4**	**108**	**97**	**14**	**0.481**	**7.02**	**5.46**	**0.77**	**36.44**	
**SD**	**13**		**3.0**	**25.8**	**24.6**	**25**	**18**	**3**	**0.303**	**3.44**	**2.98**	**0.39**	**11.93**	
**SEM**	**4**		**0.8**	**6.9**	**6.6**	**7**	**5**	**1**	**0.081**	**0.92**	**0.80**	**0.10**	**3.19**	
***t*-test** ***p*-value**	**0.034** *****		**0.503**	**0.662**	**0.941**	**<0.001** *******	**<0.001** *******	**0.001** ******	**0.368**	**<0.001** *******	**<0.001** *******	**0.033** *****	**0.502**	

**Table 2 cells-11-02358-t002:** Clinical data summary: genotype, clinical manifestations, and spirometry of patients with CF. PA, *Pseudomonas aeruginosa*, MSSA, meticillin-sensitive *Staphylococcus aureus*, MRSA, meticillin-resistant *Staphylococcus aureus*, AF, *Aspergillus fumigatus*, A. Sp., *Achromobacter* species, BCC, *Burkholderia cepacia* complex. Red shades of background color indicate the different analytical series.

Sample IDs	Genotype	Pancreatic Insufficiency Onset	Diabetes Onset	Sputum Microbiology	FEV1 %Predicted
**20_497**	**F508del/Y913S**	**NO**	**NO**	**PA/MSSA**	**44.0**
**20_504**	**F508del/F508del**	**YES**	**NO**	**mucoid PA**	**46.4**
**20_511**	**F508del/ND**	**NO**	**NO**	**mucoid PA/AF**	**83.0**
**20_1257**	**F508del/F508del**	**YES**	**NO**	**PA/A. sp.**	**39.0**
**20_1264**	**F508del/306insA**	**YES**	**NO**	**PA/A. sp.**	**63.3**
**20_1269**	**F508del/F508del**	**YES**	**YES**	**PA**	**60.0**
**20_1277**	**F508del/1717- 1G- > A**	**YES**	**NO**	**-**	**101.0**
**20_2375**	**F508del/1002-113-110delGAAT**	**NO**	**NO**	**AF/MSSA**	**127.0**
**20_2380**	**F508del/F508del**	**YES**	**NO**	**PA/AF/MSSA**	**70.0**
**20_2387**	**F508del/P574H**	**NO**	**NO**	**PA**	**73.0**
**20_2394**	**F508del/2184insA**	**YES**	**NO**	**BCC/MRSA**	**42.0**
**20_2400**	**1717-3T- > G/1717-3T- > G**	**NO**	**NO**	**-**	**94.0**
**20_2554**	**F508del/F508del**	**YES**	**NO**	**intermittent PA**	**103.0**
**20_2559**	**F508del/F508del**	**YES**	**YES**	**A. sp./MRSA**	**59.0**
				**Mean**	**71.8**
				**Median**	**66.7**
				**SD**	**26.7**
				**SEM**	**7.1**

## Data Availability

The MS proteomics data have been deposited to the ProteomeXchange Consortium (http://proteomecentral.proteomexchange.org, upload date 16 September 2021) [60] via the PRIDE partner repository [61] with the data set identifiers PXD028518 and 10.6019/PXD028518 (username: reviewer_pxd028518@ebi.ac.uk, password: 2N6leuef).

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
