# Peer review of "Sweat Proteomics in Cystic Fibrosis: Discovering Companion Biomarkers for Precision Medicine and Therapeutic Development"

_cells, 2022, doi:10.3390/cells11152358_

Round 1

Reviewer 1 Report

I appreciate the authors of Burat et al. in addressing my concerns. A few minor revisions that can be made include:

1.     I understand the point the authors are trying to make when claiming PERMANOVA  to be an unsupervised test, however this can be confusing to the reader who understands the text book definition of supervised and unsupervised learning such as myself. In unsupervised analysis there is only one data matrix, X, whereas in supervised analysis there are two, data matrix X and a data vector or matrix Y. Regardless of what the variable Y is, it is still considered a supervised approach even if Y is unrelated to the disease group. Unsupervised methods include dimension reduction approaches such as principal component analysis (PCA), t-SNE, UMAP, autoencoders and cluster analyses such k-means, whereas supervised methods include partial least squares (PLS), linear/logistic regression. I would include PERMANOVA here since the variance computation is guided by the response variable y.

2.     Define s0 used for the FDR calculation.

Reviewer 2 Report

Studying CF patients' proteomics profile is a novel idea for improving diagnostic capability. This study has successfully done this task with very interesting findings. However, the authors mainly complicated the statistical data analysis using SPLS. I would recommend this manuscript for publication after carefully revising this manuscript.

- Page 2, line 94: The last paragraph in the introduction has some of the study findings and a summary of the results. This part should not be in the introduction. Instead, a summary of the study design and process is more appealing.

-line 163: I am not sure about the relevance of desalting the sample before mass spectrometry, as they are using chromatography. Most of the saults in these samples could be washed in the LC.

-SPLS is not considered a supervised model and has never been used in a study design. If the authors are unfamiliar with that model, they should exclude it from the manuscript. 

-The simple way of presenting these data is by showing the volcano plot between the groups and highlighting the up and down-regulated proteins after excluding the isoforms in CF compared to control.

-The correlation between the significant proteomics panel and the patient phenotype is unrealistic at this stage with such a small sample size. 

-line 252, the value of the p-value cut-off needs to be assigned

-In the volcano plots they need to determine the number of up and down-regulated proteins in the figure legends, text, and tables.

-The tables in figure 3, are not clear to me mainly having the concentration of Na twice. needs to be explained in the legend and the text,

Round 2

Reviewer 2 Report

The manuscript is highly recommended for publication in the current format after clarifying the concerns

This manuscript is a resubmission of an earlier submission. The following is a list of the peer review reports and author responses from that submission.

Round 1

Reviewer 1 Report

The manuscrip from Burat et al. presents a proteomic analysis of sweat of 14 CF patients and 28 controls. The authors shows that 1057 proteins were quantifies, and almost half of them , 402, were found differentially abundant between CF and control. Then they proceed with a correlation analysis of the proteomic profiles with the clinical data, drawing conclusions on prediction of severity based on different subset of proteins. Final, a validation experiment is shown to confirm that the actin filaments are altered in the sweat of CF patients.

I find the idea of looking at the proteomic profile of the sweat in CF patients very pertinent and appealing. I have, however, major concerns about this work.

Major issues :

  1. As stated above, I agree in principle on the pertinence of analysing the sweat proteome in CF patients. However, the whole experiment has been built to find CF biomarkers by comparing CF patients to controls. In the inroductin, it is argued that CF biomarkers are needed to refine diagnosis for rare cases of “CFTR-Related Metabolic Syndrome/CF Screen Positive, Inconclusive Diagnosis (CRMS/CFSPID)”.  The real validation for this work is therefore to include such patients to prove that the CF markers identified can help their diagnosis. I suggest the authors perfom such experiments by LC-MS/MS, WB or epifluorescence.
  2. 14 CF patients are very few to draw conclusions, especially on patient subclustering. Furthermor i) the CF group is not homogenous by sex (mostly male) or genotype; ii) the authors indicate “no additional drug” for the CF patients, but they do not state which CF drug they were taking at the time of sample collection (if any). This may have an impact on the attempts of patient stratification and should be taken into account.
  3. The experiment was ran in 5 batches of sample prep and LC-MSMS analyisis. I don’t understand how the authors aim to reduce technical bias by preparing and performing LC MSMS analysis in 5 separate series, and yet keeping CF and controls in separate series. The batch effect should be carefully evaluated by variability tests,  and, if present, corrected for and taken into account in the analysis.
  4. The authors specify that 3 pool of control individuals were prepared and analysed in three separate runs each in order to build a library (used for MBR). However, this was not done with the CF samples. In the end, all samples were matched to the library. This may induce a bias in the number of CF specific proteins identified. Please explain and/or test that the main results are no altered by the absence o the library.
  5. The authors indicate that “LFQ was enabled and separated between control and CF groups..” (line 212). This may be very misleading for the quantification. According the Mat and Met, the authors injected the same amount of peptides in the LC-MS/MS, if so LFQ enabled for all CF and controls should have been applied. The authors need to prove that the separating the LFQ parameters (hence the normalization across all samples) has not induced a bias and driven up the number of proteins identified as differential.  The analysis should be run again putting the controls and the CF in the same parameter group.
  6. The volcano plot in fig 4b lacks the threshold parameters. On the material and method the authors state that “Control versus CF group comparison was achieved by a two-sample Student’s t-test and a p-value-based FDR calculation. Proteins with a FDR below 0.05 were considered significantly differently expressed between control and CF groups.”. in the supplementary table “CF-related abundance” only p-values are reported and not the q-value (as expected after FDR correction). Please explain and correct/complete.
  7. About the validation experiment on epifluorescence from the results section the reader may think that only 1 control and 1 CF were sampled (it is stated in mat and methods that the experiment was done in n=5). The 10 slides should be submitted in the main manuscript of as supplementary figures. In the legend of figure 7, please state more clearly the number of replicates both for A and B). The image in Fig 7A are very dark and hardly readable. Maybe you can adjust the contrast?
  8. I don’t see the pertinence of the interaction network of the sweat CF markers and CFTR/F508del CFTR. This figure could be eliminated, or moved in supplementary.
  9. The discussion is a bit long, it could be improved.

Author Response

Manuscript cells-1580145 – Response to reviewers’ comments and suggestions

Response to reviewer #1

We are thankful for your thorough and critical reading of our manuscript and for your insightful comments and suggestions to improve its clarity.

Comment 1: “As stated above, I agree in principle on the pertinence of analysing the sweat proteome in CF patients. However, the whole experiment has been built to find CF biomarkers by comparing CF patients to controls. In the inroductin, it is argued that CF biomarkers are needed to refine diagnosis for rare cases of “CFTR-Related Metabolic Syndrome/CF Screen Positive, Inconclusive Diagnosis (CRMS/CFSPID)”.  The real validation for this work is therefore to include such patients to prove that the CF markers identified can help their diagnosis. I suggest the authors perfom such experiments by LC-MS/MS, WB or epifluorescence.”

Response 1: We thank the reviewer for the positive comments on the pertinence of our work. As stated in the Introduction section, the aim of the study was the discovery of candidate biomarkers of potential clinical relevance for the diagnosis of CF. Importantly, we also wanted to highlight that it would be interesting to discover candidate biomarkers for CF prognosis and estimation of the disease severity as well as for therapeutic development and clinical response.

To this end, control and CF groups were compared to discover candidate CF diagnosis protein biomarkers and, for the first time, gather evidence of the relevance of sweat as a biofluid of interest for such purpose. By comparing subgroups of patients, we also gathered promising evidence pointing out that candidate biomarkers of genotype, pancreatic function, or lung function could be found in sweat.

As for CRMS/CFSPID, it was evoked to illustrate why CF diagnosis was still a challenge as of today. And while we agree it would be interesting to study patients with CRMS/CFSPID, CRMS/CFSPID was not the focus of the present study. The current work is an early discovery comparative study to identify candidate sweat biomarkers of CF in an untargeted way before further validation with targeted approaches.

The introduction was tweaked to better feature the goals of the current study according to the reviewer’s comment:

“ii) the struggle with the differential diagnosis between CF and CFTR-Related Metabolic Syndrome/CF Screen Positive, Inconclusive Diagnosis (CRMS/CFSPID), (line 61)

>> “ii) the struggle with the differential diagnosis in a small fraction (1 – 2%) of equivocal cases (e.g., between CF and CFTR-Related Metabolic Syndrome/CF Screen Positive, Inconclusive Diagnosis (CRMS/CFSPID) cases),”

“with a view to discovering new protein biomarkers for precision medicine and therapeutic development,” (line 93)

>>”with a view to discovering new protein biomarkers of CF for precision medicine and therapeutic development,”

Comment 2: “14 CF patients are very few to draw conclusions, especially on patient subclustering. Furthermor i) the CF group is not homogenous by sex (mostly male) or genotype; ii) the authors indicate “no additional drug” for the CF patients, but they do not state which CF drug they were taking at the time of sample collection (if any). This may have an impact on the attempts of patient stratification and should be taken into account.”

Response 2: We agree the number of patients in a cohort is of utmost importance from a statistical standpoint, depending on what the study aims at demonstrating. The current study has been designed as an early biomarker discovery study, and the size of the cohort is in good agreement with this kind of proof-of-concept and feasibility studies. Observations about the patient sub-clustering were limited to the patient stratification according to proteome profiles, as were the subsequent conclusions. While we agree the CF group is not homogenous for sex, we compromised on the patient recruitment based on previous results from our group (https://doi.org/10.3390/ijms221910871) showing there was no inter-gender variability of the global sweat proteome. We concur with the reviewer that, under the knowledge that more than 2,000 putative CFTR mutations were identified and that consistent variability in the severity of CF was observed even in homozygotic twins, CF is far from being a homogeneous condition. However, attention has been paid to recruiting size-equivalent CF subgroups related to the F508del genotype (homozygous and heterozygous) and conclusions were limited to this mutation.

We thank the reviewer for the remarks concerning the medication of patients at the time of collection. Exclusion criteria for recruitment of patients were to be involved in any other clinical trials and to be under treatment with any drug therapy modulating CFTR function and/or expression (corrector/potentiator therapy). To address the reviewer’s comment, the following has been added to the Experimental design and statistical rationale section: “Patients with CF, tested under stable conditions, were not enrolled in other clinical trials or under CFTR modulator therapies.” (lines 299-300).

Comment 3: “The experiment was ran in 5 batches of sample prep and LC-MSMS analyisis. I don’t understand how the authors aim to reduce technical bias by preparing and performing LC MSMS analysis in 5 separate series, and yet keeping CF and controls in separate series. The batch effect should be carefully evaluated by variability tests,  and, if present, corrected for and taken into account in the analysis.”

Response 3: We understand the reviewer’s concern about technical biases which is a crucial point. We previously showed the variability between series of the same group was negligible compared to biological variability (https://doi.org/10.3390/ijms221910871). This is now explained in the manuscript along with the aforementioned citation.

To address the reviewer’s comment, the following sentence has been added to the Experimental design and statistical rationale section: “The inter-series technical variability for a given group was negligible when compared to the biological variability [19].” (lines 312-314).

Comment 4: “The authors specify that 3 pool of control individuals were prepared and analysed in three separate runs each in order to build a library (used for MBR). However, this was not done with the CF samples. In the end, all samples were matched to the library. This may induce a bias in the number of CF specific proteins identified. Please explain and/or test that the main results are no altered by the absence o the library.”

Response 4: We agree the ideal library design would have been a mixture of all samples. However, when comparing the control and CF sweat proteomes, it was found that they were highly similar (Fig. 2b, 98.3 % of proteins in common), “exclusive” proteins being found in low abundance in a low number of samples (peptides/proteins near the limit of detection). And so, when considering the low number and volume of available CF samples, we compromised by limiting the library to control samples, saving CF samples for other analyses. Besides, another element to account for is the matching between samples which is as important if not more important than sample-library matching for identification performance, according to optimization tests.

To address the reviewer’s comment, the following test has been added to the Experimental design and statistical rationale section: “For the same reasons, no CF sample library was analyzed considering the high similarity between the control and CF proteome as well as the enabled inter-sample MbR.” (lines 308-310).

Comment 5: “The authors indicate that “LFQ was enabled and separated between control and CF groups..” (line 212). This may be very misleading for the quantification. According the Mat and Met, the authors injected the same amount of peptides in the LC-MS/MS, if so LFQ enabled for all CF and controls should have been applied. The authors need to prove that the separating the LFQ parameters (hence the normalization across all samples) has not induced a bias and driven up the number of proteins identified as differential.  The analysis should be run again putting the controls and the CF in the same parameter group.”

Response 5: Two raw data analyses were conducted in parallel: one with LFQ separation between the control and CF parameter groups (as presented in the paper) and one without LFQ separation. Our choice was made based on two elements:

- the comparison of Volcano plots between i) separated LFQ or, ii) non-separated LFQ normalized data and, iii) non-normalized data.

- the quantification of spiked-in standard proteins from MPDS Mix1 protein digest with i) separated LFQ, ii) non-separated LFQ normalized data or, iii) non-normalized data.

Considering an equivalent amount of proteins injected for all samples, the main argument to choose the normalization to apply was that no LFQ normalization and separated LFQ normalization led to the same results in protein quantification for both the proteomic differential study (same conclusions can be drawn notwithstanding less significant protein abundance variations with no LFQ normalization) and the quality control evaluation based on the standard digest spike-in.

On the contrary, non-separated LFQ normalization led to a quantification bias when compared to non-normalized data (e.g., MPDS Mix1 proteins are in a 1:1:1:1 ratio that was lost with non-separated LFQ normalization). This observation proved that the protein abundances in each group were too different to apply a global normalization.

So, we opted for a LFQ normalization by group to reduce inter-sample variability within each group while leading to the same conclusions we would get with no LFQ normalization (which could be used instead because an equivalent amount of proteins was injected for all samples), control and CF samples being in two separate parameter groups. Details about this point were added to improve the manuscript.

To address the reviewer’s comment, the following sentence has been added to the Experimental design and statistical rationale section: “Inter-sample technical variability was reduced by separating LFQ normalization between the control and CF groups: considering the equivalent amount of proteins injected for all samples, the computation of both non-normalized data and separated LFQ normalized data drew the same conclusions in protein quantification (for both differential proteomic analysis and quantification quality control via the standard digest spike-in). However, non-separated LFQ normalization (normalized data not provided) led to a quantification bias (e.g., the 1:1:1:1 ratio of the standard protein digest spike-in was lost). Protein abundances between the two groups were not suitable for a global normalization.” (lines 314-321).

Comment 6: “The volcano plot in fig 4b lacks the threshold parameters. On the material and method the authors state that “Control versus CF group comparison was achieved by a two-sample Student’s t-test and a p-value-based FDR calculation. Proteins with a FDR below 0.05 were considered significantly differently expressed between control and CF groups.”. in the supplementary table “CF-related abundance” only p-values are reported and not the q-value (as expected after FDR correction). Please explain and correct/complete.”

Response 6: We thank the reviewer for this comment. The text has been corrected as follows “Control versus CF group comparison was achieved by a two-sample Student’s t-test with a p-value-based threshold. Proteins with a p-value below 0.05 were considered significantly differently expressed between control and CF groups.” (lines 236-238).

Concerning the comment on q-value, no q-value was reported in the supplementary table “CF-related abundance” because no q-value was computed for this t-test (as performed by Perseus).

To address the reviewer’s comment, the following sentence: “Control versus CF Volcano plot visualization was achieved by a two-sample Student’s t-test with a permutation-based FDR calculation (test=t-test; side=both; number of randomizations=250; no grouping in randomizations; FDR=0.05; s0=0.1).” has been added to the Material and Methods section (lines 242-244). Figure 4b has been modified to address the threshold criteria.

Comment 7: About the validation experiment on epifluorescence from the results section the reader may think that only 1 control and 1 CF were sampled (it is stated in mat and methods that the experiment was done in n=5). The 10 slides should be submitted in the main manuscript of as supplementary figures. In the legend of figure 7, please state more clearly the number of replicates both for A and B). The image in Fig 7A are very dark and hardly readable. Maybe you can adjust the contrast?

Response 7: We thank the reviewer for this comment on the characterization of sweat actin by epifluorescence microscopy, and the manuscript has been completed accordingly. Micrographs of Figure 7 are specimens for the 100 snapshots taken for each condition (5 controls or 5 CFs times 20 micrographs) and we considered adding 100 snapshots as supplementary materials would have been a bit excessive. The specimen micrographs are raw and already well contrasted to see the microfilaments. The control micrograph appears dark because it is mostly empty with only a few short microfilaments. Besides, post- manipulation of images is not a recommended practice.

To address the reviewer’s comment, the following sentence has been added to the Materials and Methods section “For each individual microscope slide, 20 snapshots were taken as followed: 10 snapshots at an operator-chosen position and 10 snapshots at a random position.” was added to the Materials and Methods section (lines 284-285).

To address the reviewer’s comment, the following text has been added to the Figure 7 legend: “(n=5, 5 controls compared to 5 patients with CF; nmicrographs=20 per individual)”.

Comment 8: “I don’t see the pertinence of the interaction network of the sweat CF markers and CFTR/F508del CFTR. This figure could be eliminated, or moved in supplementary.”

Response 8: We are sorry the reviewer does not seem to recognize the pertinence of unveiling the protein interactomes of CFTR and F508del CFTR. Numerous recent studies have argued in favor of the fact that dissecting protein-protein interaction networks can lead to a better understanding of cellular processes and pathways with temporal and spatial precision. We are convinced that shedding light on the dynamic interactions of CFTR with other proteins help to understand how sweat gland epithelial cells adjust their responses to different stimuli and how the presence of a mutation such as F508del, provoking cell stress responses and associated to severe disease phenotypes, impacts on these cellular networks and on differential protein abundances in CF sweat. We do believe, as stated in the Discussion section that “… from a clinical standpoint, sweat CFTR interactors and F508del CFTR interactors in CF-specific abundance are of utmost interest in the search for biomarkers of phenotypic rescue to benefit new therapeutic developments. Accordingly, we have decided not to remove figure 6 from the main text section and not to move it to the Supplemental Materials section.

Comment 9: “The discussion is a bit long, it could be improved.”

Response 9: All points discussed in the article were necessarily addressed to parallel the diversity of original findings provided in our work. The length of the discussion reflected the dual nature of the study novelties: on one hand, the discussion about the proteomic profiling of CF sweat had to address the clinical and pathophysiological implications of the discovered biomarkers, on the other hand, the discussion about sweat actin had to address the greater mechanistical and pathophysiological implications and propose a hypothetical pathophysiological mechanism model. All in all, to be the most comprehensive for the diversity of readers of this Cells journal Special Issue, from clinicians with a focus on the patients to cellular/molecular biologists with a more fundamental interest, the discussion ended up an adequate length to address all points of interest, getting the most out of the results. The Discussion section is dense without digression, and it cannot be shortened without compromising the understanding of original findings through the essential aforementioned points.

Reviewer 2 Report

Burat et al. present a non-invasive sweat profiling procedure to identify protein biomarkers of cystic fibrosis. Given the number of statistical comparisons made in this study, it is possible that many of these findings may be due to spurious correlations. I provide some advice on improving the statistical modelling used in this study to reduce the number of false discoveries that may have been identified.

Major:

  1. Despite there being no statistical differences in the collected volume between groups, normalizing all samples such that the average collected volume is the same across samples may affect the downstream differential expression analysis results since protein levels in each sample will be shifted in different directions (increased/decreased). The normalization factor for a given sample can be computed as a ratio between the average collected volume for that sample divided by the average collected volume across all samples. Then multiplying all values of that sample by this factor will adjust for difference in collected volumes between samples.
  2. The PERMANOVA is useful in identifying outcome variables (y) and might explain the variability in the proteomics dataset (X), however since multiple outcomes variables were tested it might also be useful to employ a methodology that capture the correlation between outcomes variables. Partial Least Squares (PLS) allows for exploring the association between matrices of independent (X, protein data) and dependent variables (Y, clinical data) by maximizing the correlation between principal components from X and Y. Sparse PLS will impose variable selection so that one can interpret which variables in X and Y are associated with one another in the context of the entire dataset. See here for a tutorial on PLS and sPLS ().
  3. It is unclear if an FDR was used to correct for multiple testing correction when testing for differential abundance. FDR is only mentioned in the methods for differential abundance testing with a cut-off of 5%. However, based on the volcano plot it is difficult to determine if those cut-offs (solid black line) depict a p-value threshold or FDR threshold. Please state the type of FDR used (Benjamini Hochberg, permutation-based FDR). I would recommend using limma R-library (Ritchie ME, Phipson B, Wu D, Hu Y, Law CW, Shi W, Smyth GK (2015). “limma powers differential expression analyses for RNA-sequencing and microarray studies.” Nucleic Acids Research43(7), e47. doi: 1093/nar/gkv007.) which performs  moderated t-tests in order to account for the differences in the scale of variances (some protein will have very small variances and some will have very large).
  4. What is the rationale for computing AUCs when you have already found that a given protein is differentially expressed?  I am not sure surprised that the AUCs ~ 1. I would be surprised if they were not. If you are interested in identifying a set of protein that are predictive of CF, please apply some classification methods and assess their performance using cross-validation using all the proteins (NOT the differentially expressed ones). I recommend the use of the caret R-package which contains over 230 machine learning algorithms such as Elastic Net regularized regression models that simultaneously performs model fitting and variable selection. I highly recommended reading Chapter 7: Model Assessment and Selection of the Elements of Statistical Learning (http://statweb.stanford.edu/~tibs/ElemStatLearn/). In the absence of an external independent dataset, the performance of biomarkers should be estimated using cross-validation.

Minor:

  1. The use of “n=X” is used as a convention to refer to as “number of observations” which can be patients, assays, samples etc. On the other hand, p=X is used to refer to the “number of variables” such as proteins, genes etc. From lines 95-97 n is used to refer to as the number of proteins which can be confusing. Please replace n with p to easy reading of the text.
  2. Table 1 gradient colouring is confusing. Why is the table of subjects also highlighted in green and red and what does the gradient in the colours mean (light green, intermediate green, dark green)? Also I am not sure why Table 2 also has a gradient of light and dark red?
  3. Figures: improve readability of figures:
    1. Figure 4B: change y-axis label to -log10(p-value), label what the solid black lines mean (p=0.05 threshold I am assuming). Instead using an FDR threshold of 5% for example.
    2. For Figure 4C which depicts the barplots for multiple proteins, order the proteins based on average expression so that it is easier to interpret. For example for Figure 4Cb arrange as GPT, EML4, MDH2, ARG1 from least to highest average expression.
  4. I do not see the relevance of identifying subject clusters for the hierarchical clustering performance since this is performed on all samples and variables. It is to me a good quality control metric to remove samples as done in Supplementary Figure 1. However, if the authors are really interested in subphenotypes they are welcome to try the ConsensusClusterPlus R-library which provides information about cluster stability (Wilkerson, D. M, Hayes, Neil D (2010). “ConsensusClusterPlus: a class discovery tool with confidence assessments and item tracking.” Bioinformatics26(12), 1572-1573). Internal cluster validation should be explored using the silhouette index. A statistical test to compare the data distribution between the clusters should be performed using the sigclust Test (Liu, Yufeng, Hayes, David Neil, Nobel, Andrew and Marron, J. S, 2008, Statistical Significance of Clustering for High-Dimension, Low-Sample Size Data, Journal of the American Statistical Association 103(483) 1281-1293). This will provide evidence into whether these clusters are invariant to permutations of the samples and are actually meaningful.
  5. It is also unclear how imputation of missing values was conducted for differential abundance analysis, did you just ignore NAs?

Author Response

Manuscript cells-1580145 – Response to reviewers’ comments and suggestions

Response to reviewer #2

We are thankful for your thorough and critical reading of our manuscript and for your insightful comments and suggestions to improve its overall quality.

Major revisions:

Comment 1: “Despite there being no statistical differences in the collected volume between groups, normalizing all samples such that the average collected volume is the same across samples may affect the downstream differential expression analysis results since protein levels in each sample will be shifted in different directions (increased/decreased). The normalization factor for a given sample can be computed as a ratio between the average collected volume for that sample divided by the average collected volume across all samples. Then multiplying all values of that sample by this factor will adjust for difference in collected volumes between samples.”

Response 1: We thank the reviewer for this comment. We agree a normalization according to sweat volumes would generate a quantification and proteome comparison bias as the total protein concentration is not constant across samples. For that reason, the experimental design of the study did not include such a normalization. The comparison of the control and CF proteome was performed after normalization of the amount of proteins between samples (10 µg at the start of the sample prep, 3 µg injected for each sample).

Comment 2: “The PERMANOVA is useful in identifying outcome variables (y) and might explain the variability in the proteomics dataset (X), however since multiple outcomes variables were tested it might also be useful to employ a methodology that capture the correlation between outcomes variables. Partial Least Squares (PLS) allows for exploring the association between matrices of independent (X, protein data) and dependent variables (Y, clinical data) by maximizing the correlation between principal components from X and Y. Sparse PLS will impose variable selection so that one can interpret which variables in X and Y are associated with one another in the context of the entire dataset. See here for a tutorial on PLS and sPLS ().”

Response 2: As the reviewer pointed out, PERMANOVA was the multivariate test of choice to identify relevant outcome variables among available clinical parameters. In the context of an early biomarker discovery study, the aim was to assess, in an unsupervised manner, the potential clinical relevance of global sweat proteome profiles in CF diagnosis and prognosis, and this is what we aimed at. Although PLS and sPLS are interesting propositions, after some tests, we do not see the added value in light of our hypothesis, when compared with the PERMANOVA test and other unsupervised and supervised approaches we used. No change in the manuscript has been made to address this comment.

Comment 3: “It is unclear if an FDR was used to correct for multiple testing correction when testing for differential abundance. FDR is only mentioned in the methods for differential abundance testing with a cut-off of 5%. However, based on the volcano plot it is difficult to determine if those cut-offs (solid black line) depict a p-value threshold or FDR threshold. Please state the type of FDR used (Benjamini Hochberg, permutation-based FDR). I would recommend using limma R-library (Ritchie ME, Phipson B, Wu D, Hu Y, Law CW, Shi W, Smyth GK (2015). “limma powers differential expression analyses for RNA-sequencing and microarray studies.” Nucleic Acids Research, 43(7), e47. doi: 1093/nar/gkv007.) which performs  moderated t-tests in order to account for the differences in the scale of variances (some protein will have very small variances and some will have very large).”

Response 3: We thank the reviewer for this comment. To address the reviewer’s comment on differential protein quantification, the text has been modified as follows: “Control versus CF group comparison was achieved by a two-sample Student’s t-test with a p-value-based threshold. Proteins with a p-value below 0.05 were considered significantly differently expressed between control and CF groups.” (lines 236-238).

No q-value was reported in the supplementary table “CF-related abundance” because no q-value was computed for this t-test (Perseus doesn’t compute q-value as part of the Volcano plot results).

To adequately address the reviewer’s comment on the volcano plot, the following sentence has been added to the Material and Methods section: “Control versus CF Volcano plot visualization was achieved by a two-sample Student’s t-test with a permutation-based FDR calculation (test=t-test; side=both; number of randomizations=250; no grouping in randomizations; FDR=0.05; s0=0.1).” (lines 242-244).

Accordingly, Figure 4b has been modified to address the threshold criteria.

Comment 4: “What is the rationale for computing AUCs when you have already found that a given protein is differentially expressed?  I am not sure surprised that the AUCs ~ 1. I would be surprised if they were not. If you are interested in identifying a set of protein that are predictive of CF, please apply some classification methods and assess their performance using cross-validation using all the proteins (NOT the differentially expressed ones). I recommend the use of the caret R-package which contains over 230 machine learning algorithms such as Elastic Net regularized regression models that simultaneously performs model fitting and variable selection. I highly recommended reading Chapter 7: Model Assessment and Selection of the Elements of Statistical Learning (http://statweb.stanford.edu/~tibs/ElemStatLearn/). In the absence of an external independent dataset, the performance of biomarkers should be estimated using cross-validation.”

Response 4: We thank the reviewer for this comment.

The discovery of potential candidate biomarkers was performed by applying highly stringent cut-offs (difference, p-value, occurrence) to the set of differentially abundant proteins. These candidate biomarkers have potential clinical relevance that must be tested and validated with larger cohorts and adapted statistical models (as you suggested) in the future.

The computation of AUC had two purposes:

- a rapid ranking of those high potential candidate biomarkers between good (0.8<r<0.9) and excellent (r>0.9) discrimination performance

- the easier readability of the results by clinicians that might not have been used to interpret proteomic results in another form.

Minor revisions:

Comment 1: “The use of “n=X” is used as a convention to refer to as “number of observations” which can be patients, assays, samples etc. On the other hand, p=X is used to refer to the “number of variables” such as proteins, genes etc. From lines 95-97 n is used to refer to as the number of proteins which can be confusing. Please replace n with p to easy reading of the text.”

Response 1: This is a good point. The manuscript has been corrected to be clearer.

Comment 2: “Table 1 gradient colouring is confusing. Why is the table of subjects also highlighted in green and red and what does the gradient in the colours mean (light green, intermediate green, dark green)? Also I am not sure why Table 2 also has a gradient of light and dark red?”

Response 2: We thank the reviewer for this comment. Green was meant to highlight control subjects while red highlighted patients with CF. Shades were used to highlight the distribution of individuals between the different analytical series. The table legends have been completed with the required information.

Comment 3: “Figures: improve readability of figures:

- Figure 4B: change y-axis label to -log10(p-value), label what the solid black lines mean (p=0.05 threshold I am assuming). Instead using an FDR threshold of 5% for example.

- For Figure 4C which depicts the barplots for multiple proteins, order the proteins based on average expression so that it is easier to interpret. For example for Figure 4Cb arrange as GPT, EML4, MDH2, ARG1 from least to highest average expression.”

Response 3: Thank you to the reviewer for the comments. The figures have been modified accounting for these remarks to improve readability.

Comment 4: “I do not see the relevance of identifying subject clusters for the hierarchical clustering performance since this is performed on all samples and variables. It is to me a good quality control metric to remove samples as done in Supplementary Figure 1. However, if the authors are really interested in subphenotypes they are welcome to try the ConsensusClusterPlus R-library which provides information about cluster stability (Wilkerson, D. M, Hayes, Neil D (2010). “ConsensusClusterPlus: a class discovery tool with confidence assessments and item tracking.” Bioinformatics, 26(12), 1572-1573). Internal cluster validation should be explored using the silhouette index. A statistical test to compare the data distribution between the clusters should be performed using the sigclust Test (Liu, Yufeng, Hayes, David Neil, Nobel, Andrew and Marron, J. S, 2008, Statistical Significance of Clustering for High-Dimension, Low-Sample Size Data, Journal of the American Statistical Association 103(483) 1281-1293). This will provide evidence into whether these clusters are invariant to permutations of the samples and are actually meaningful.”

Response 4: We thank the reviewer for providing known references that would certainly be valuable for sub-clustering and sub-phenotype analyses. However, more than identifying subject clusters, the aim was to identify sweat proteome profile clusters. This was mandatory for the application of the PERMANOVA test to the comparison between sweat proteome profiles and the distribution of clinical data values. In the context of an early biomarker discovery study, it was meant to evaluate the potential clinical relevance of sweat proteome profiles. We consider that our project having been designed to identify sweat proteome profile clusters and not to establish comparisons between subject subclusters, would not be subject to applications of the ConsensusClusterPlus R-library and of the sigclust test. We are also convinced that for such applications in the field of a non-homogeneous genetic disease such as Cystic Fibrosis, much larger size cohorts would be required. We do believe that applying PERMANOVA for comparisons between sweat proteome profiles and distribution of clinical data values, as carried out in our work, is in adequacy with the context of an early biomarker discovery study designed with the goal to evaluate the potential clinical relevance of CF-related sweat proteome profiles.

Comment 5: “It is also unclear how imputation of missing values was conducted for differential abundance analysis, did you just ignore NAs?”

Response 5: NAs are essentially ignored but differential protein quantification was performed on proteins having at least 50% of valid values in one group. Therefore, we did not apply any imputation of missing values.

Round 2

Reviewer 1 Report

The authors invested time and efforts in their answers, and some of my concerns have been addressed through clarification and correction of the text.

Howewer, some major concerns remain:

Comment 5

In comment 4, the authors claim that “when comparing the control and CF sweat proteomes, it was found that they were highly similar (Fig. 2b, 98.3 % of proteins in common), “exclusive” proteins being found in low abundance in a low number of samples”, to justify the use of a partial library. Yet, in answer to comment 5 they go at great length about the need to use separated LFQ parameter groups because of variability “we opted for a LFQ normalization by group to reduce inter-sample variability within each group”. They claim that this leads to  “the same conclusions we would get with no LFQ normalization” .

I am not convinced, as this is not logical.

Nonetheless , I am open to evaluate this explanation if supported by data the authors mention :

-please provide the volcano plots for the three conditions  (no LFQ, LFQ global, LFQ by group)

-please provide the ratio of the standard protein digest spike-in used as control (and how it deviates form 1:1:1:1 in the three data analysis conditions).

Comment 6

FDR correction must be applied, a simple threshold by p-value is not correct.

“Concerning the comment on q-value, no q-value was reported in the supplementary table “CF-related abundance” because no q-value was computed for this t-test (as performed by Perseus).”. Perseus CAN compute the q-value if you perform a ttest using the same parameters as you used to generate the volcano plot.

Comment 8 (minor)

If so pertinent, then it need to better explained or introduced.

As I understand, the authors map a physical protein-protein interaction network into sweat CF differentially abundant proteins to create a CFTR (physical? Functional?) protein network.

Reviewer 2 Report

Reviewer Response 1: Can the authors please reference the line numbers as to where amount of proteins between samples (10 µg at the start of the sample prep, 3 µg injected for each sample) is mentioned in the manuscript draft?

Reviewer Response 2: Can the authors please elaborate on which tests were conducted in order evaluate that PLS/sPLS have “no added value”? I do not consider PERMANOVA as an unsupervised approach as the purpose of this method is to determine differences in data distribution between pre-defined response variable (as demonstrated by the p-values in Figure 3). A p-value adjustment for multiple testing should also be applied here.

Reviewer Response 3: A nominal p-value cut-off should be considered even in the context of a pilot study because the p-value is a random variable and spurious correlations can arise. An FDR correction should be applied because it will quantify the number of false discoveries that may exist for a given threshold. A p-value histogram should also be considered to demonstrate that the p-value distribution is different from the null (from that of randomly generated data); see details in this blog post (http://varianceexplained.org/statistics/interpreting-pvalue-histogram/).

Reviewer Response 4: The authors are welcome to use AUCs but they have been used incorrectly. The authors need to report the cross-validated AUCs not computing them directly after differential expression analysis. Please apply some classification methods and assess their performance using cross-validation using all the proteins (NOT the differentially expressed ones). I recommend the use of the caret R-package which contains over 230 machine learning algorithms such as Elastic Net regularized regression models that simultaneously performs model fitting and variable selection. I highly recommended reading Chapter 7: Model Assessment and Selection of the Elements of Statistical Learning (http://statweb.stanford.edu/~tibs/ElemStatLearn/).